# Modular Lifelong Reinforcement Learning via Neural Composition

**Jorge A. Mendez**[1†], **Harm van Seijen**[2], and **Eric Eaton**[1]
[1]Department of Computer and Information Science
University of Pennsylvania
{mendezme,eeaton}@seas.upenn.edu

[2]Microsoft Research
harm.vanseijen@microsoft.com

## Abstract

Humans commonly solve complex problems by decomposing them into easier subproblems and then combining the subproblem solutions. This type of compositional reasoning permits reuse of the subproblem solutions when tackling future tasks that share part of the underlying compositional structure. In a continual or lifelong reinforcement learning (RL) setting, this ability to decompose knowledge into reusable components would enable agents to quickly learn new RL tasks by leveraging accumulated compositional structures. We explore a particular form of composition based on neural modules and present a set of RL problems that intuitively admit compositional solutions. Empirically, we demonstrate that neural composition indeed captures the underlying structure of this space of problems. We further propose a compositional lifelong RL method that leverages accumulated neural components to accelerate the learning of future tasks while retaining performance on previous tasks via off-line RL over replayed experiences.

## 1 Introduction

Reinforcement learning (RL) has achieved impressive success at complex tasks, from mastering the game of Go (Silver et al., 2016; 2017) to robotics (Gu et al., 2017; Andrychowicz et al., 2020). However, this success has been mostly limited to solving a single problem given enormous amounts of experience. In contrast, humans learn to solve a myriad of tasks over their lifetimes, becoming better at solving them and faster at learning them over time. This ability to handle diverse problems stems from our capacity to accumulate, reuse, and recombine perceptual and motor abilities in different manners to handle novel circumstances. In this work, we seek to endow artificial agents with a similar capability to solve RL tasks using *functional compositionality* of their knowledge.

In lifelong RL, the agent faces a sequence of tasks and must strive to transfer knowledge to future tasks and avoid forgetting how to solve earlier tasks. We formulate the novel problem of lifelong RL of functionally compositional tasks, where tasks can be solved by recombining modules of knowledge in various ways. While temporal compositionality has long been studied in RL, such as in the options framework, the type of *functional* compositionality we study here has not been explored in depth, especially not in the more realistic lifelong learning setting. Functional compositionality involves a decomposition into subproblems, where the outputs of one subproblem become inputs to others. This moves beyond standard temporal composition to functional compositions of layered perceptual and action modules, akin to programming where functions are used in combination to solve different problems. For example, a typical robotic manipulation solution interprets perceptual inputs via a sensor module, devises a path for the robot using a high-level planner, and translates this path into motor controls with a robot driver. Each of these modules can be used in other combinations to handle a variety of problems. We present a new method for continually training deep modular architectures, which enables efficient learning of the underlying compositional structures.

The modular lifelong RL agent will encounter a sequence of compositional tasks, and must strive to solve them as quickly as possible. Our proposed solution separates the learning process into three stages. First, the learner discovers how to combine its existing modules to solve the current task to the best of its abilities by interacting with the environment with various module combinations. Next, the agent accumulates additional information about the current task via standard RL training with

---

†Part of this work was carried out while the author was at Microsoft Research.

the optimal module combination. Finally, the learner incorporates any newly discovered knowledge from the current task into existing modules, making them more suitable for future learning. We demonstrate that this separation enables faster training and avoids forgetting, even though the agent is not allowed to revisit earlier tasks for further experience. Our main contributions include:

1. We **formally define the lifelong compositional RL problem** as a compositional problem graph, encompassing both zero-shot generalization and fast adaptation to new compositional tasks.
2. We propose two **compositional evaluation domains**: a discrete 2-D domain and a realistic robotic manipulation suite, both of which exhibit compositionality at multiple hierarchical levels.
3. We create the **first lifelong RL algorithm for functionally compositional structures** and show empirically that it learns meaningful compositional structures in our evaluation domains. While our evaluations focus on explicitly compositional RL tasks, the concept of functional composition is broadly applicable and could be used in future solutions to general lifelong RL.
4. We propose to use **batch RL** techniques for avoiding catastrophic forgetting in a lifelong setting and show that this approach is superior to existing lifelong RL methods.

## 2 RELATED WORK

**Lifelong or continual learning**    Most work in lifelong learning has focused on the supervised setting, and in particular, on avoiding catastrophic forgetting. This has typically been accomplished by imposing data-driven regularization schemes that discourage parameters from deviating far from earlier tasks' solutions (Zenke et al., 2017; Li & Hoiem, 2017; Ritter et al., 2018), or by replaying real (Lopez-Paz & Ranzato, 2017; Nguyen et al., 2018; Chaudhry et al., 2019; Aljundi et al., 2019) or hallucinated (Achille et al., 2018; Rao et al., 2019; van de Ven et al., 2020) data from earlier tasks during the training of future tasks. Other methods have instead aimed at solving the problem of model saturation by increasing the model capacity (Yoon et al., 2018; Li et al., 2019; Rajasegaran et al., 2019). A few works have addressed lifelong RL by following the regularization (Kirkpatrick et al., 2017) or replay (Isele & Cosgun, 2018; Rolnick et al., 2019) paradigms from the supervised setting, exacerbating the stability-plasticity tension. Others have instead proposed multi-stage processes whereby the agent first transfers existing knowledge to the current task and later incorporates newly obtained knowledge into a shared repository (Schwarz et al., 2018; Mendez et al., 2020). We follow the latter strategy for exploiting and subsequently improving accumulated modules over time.

**Compositional supervised learning**    While deep nets in principle enable learning arbitrarily complex task relations, monolithic agents struggle to find such complex relations given limited data. Compositional multi-task learning (MTL) methods use explicitly modular deep architectures to capture compositional structures that arise in real problems. Such approaches either require the compositional structure to be provided (Andreas et al., 2016; Hudson & Manning, 2018) or automatically discover the structure in a hard (Rosenbaum et al., 2018; Alet et al., 2018; Chang et al., 2019) or soft (Kirsch et al., 2018; Meyerson & Miikkulainen, 2018) manner. A handful of approaches have been proposed that operate in the lifelong setting, under the assumption that each component can be fully learned by training on a single task and then reused for other tasks (Reed & de Freitas, 2016; Fernando et al., 2017; Valkov et al., 2018). Unfortunately, this is infeasible if the agent has access to little data per task. Recent work proposed a framework for lifelong supervised compositional learning (Mendez & Eaton, 2021), similar in high-level structure to our proposed method for RL.

Typical evaluations of compositional learning use standard benchmarks such as ImageNet or CIFAR-100. While this enables fair performance comparisons, it fails to give insight about the ability to find meaningful compositional structures. Some notable exceptions exist for evaluating compositional generalization in supervised learning (Bahdanau et al., 2018; Lake & Baroni, 2018; Sinha et al., 2020). We extend the ideas of compositional generalization to RL, and introduce the separation of zero-shot compositional generalization and fast adaptation, which is particularly relevant in RL.

**Compositional RL**    A handful of works have considered functionally compositional RL, either assuming full knowledge of the correct compositional structure of the tasks (Devin et al., 2017) or automatically learning the structure simultaneously with the modules themselves (Goyal et al., 2021; Mittal et al., 2020; Yang et al., 2020). We propose a method that can handle both of these settings. Two main aspects distinguish our work from existing approaches: 1) our method works in a lifelong setting, where tasks arrive sequentially and the agent is not allowed to revisit previous tasks, and 2) we evaluate our models on tasks that are explicitly compositional at multiple hierarchical levels.

A closely related problem that has long been studied in RL is that of temporally extended actions, or options, for hierarchical RL (Sutton et al., 1999; Bacon et al., 2017). Crucially, the problem we consider here differs in that the functional composition occurs at every time step, instead of the temporal chaining considered in the options literature. These two orthogonal dimensions capture real settings in which composition could improve the RL process. Appendix A contains a deeper discussion of these connections. Other forms of hierarchical RL have learned state abstractions (Dayan & Hinton, 1993; Dietterich, 2000; Vezhnevets et al., 2017; Abel et al., 2018). While these are also related, they have mainly focused on a two-layer abstraction, where the policy passes the output of a state abstraction module as input to an action module. Instead, we consider arbitrarily many layers of abstraction that help the learning of both state and action representations. One additional form of composition in RL uses full task policies as the components, and combines policies together when solving a new task (e.g., by summing their value functions; Todorov, 2009; Barreto et al., 2018; Haarnoja et al., 2018; Van Niekerk et al., 2019; Nangue Tasse et al., 2020). Instead, we separate each policy itself into components, such that these components combine to form full policies.

**Batch RL** RL from fixed data introduces a shift between the data distribution and the learned policy, which strains the capabilities of standard methods (Levine et al., 2020). Recent techniques constrain the departure from the data distribution (Fujimoto et al., 2019b; Laroche et al., 2019; Kumar et al., 2019). This issue closely connects to lifelong RL: obtaining backward transfer requires modifying the policy for earlier tasks without extra experience. We exploit this connection by storing a portion of the data collected on each task and replaying it for batch RL after future tasks are trained.

## 3 THE PROBLEM OF LIFELONG FUNCTIONAL COMPOSITION IN RL

Complex RL problems can often be divided into easier subproblems. Recent years have taught us that, given unlimited experience, artificial agents can tackle such complex problems without any sense of their compositional structures (Silver et al., 2016; Gu et al., 2017). However, discovering the underlying subproblems and learning to solve those would enable learning with substantially less experience, especially when faced with numerous tasks that share common structure. While our work shares this premise with hierarchical RL efforts, those works do not consider the formulation we present here, where *functional* composition occurs at multiple hierarchical levels of abstraction.

Formally, an RL problem is given by a Markov decision process (MDP) $\mathcal{Z} = \langle \mathcal{S}, \mathcal{A}, R, T, \gamma \rangle$, where $\mathcal{S}$ is the set of states, $\mathcal{A}$ is the set of actions, $T : \mathcal{S} \times \mathcal{A} \mapsto \mathcal{S}$ is the probability distribution $P(\boldsymbol{s'} \mid \boldsymbol{s}, \boldsymbol{a})$ of transitioning to state $\boldsymbol{s'}$ upon executing action $\boldsymbol{a}$ in state $\boldsymbol{s}$, $R : \mathcal{S} \times \mathcal{A} \mapsto \mathbb{R}$ is the reward function measuring the goodness of a given state-action pair, and $\gamma$ is the discount factor that reduces the importance of rewards obtained far in the future. The agent's behavior is dictated by a (possibly stochastic) policy $\pi : \mathcal{S} \mapsto \mathcal{A}$ that selects the action to take at each state. The goal of the agent is to find the policy $\pi^* \in \Pi$ that maximizes the discounted long-term returns $\mathbb{E}[\sum_{i=0}^{\infty} \gamma^i R(\boldsymbol{s}_i, \boldsymbol{a}_i)]$.

An RL problem $\mathcal{Z}$ is a composition of subproblems $F_1, F_2, \ldots$ if its optimal policy $\pi^*$ can be constructed by combining solutions to those subproblems: $\pi^*(\boldsymbol{s}) = m_1 \circ m_2 \circ \cdots$, where each $m_i \in \mathcal{M} : \mathcal{X}_i \mapsto \mathcal{Y}_i$ is the solution to the corresponding $F_i$. We first consider these subproblems from an intuitive perspective, and later define them precisely. In RL, each subproblem could involve pure sensing, pure acting, or a combination of both. For instance, in our earlier robotic manipulation example, the task can be decomposed into recognizing objects (sensing), detecting the target object and devising a plan to reach it (combined), and actuating the robot to grasp the object (acting).

In lifelong compositional RL, the agent will be faced with a sequence of MDPs $\mathcal{Z}^{(1)}, \ldots, \mathcal{Z}^{(T_{\max})}$ over its lifetime. We assume that all MDPs are compositions of different subsets from $k$ shared subproblems $\mathcal{F} = \{F_1, \ldots, F_k\}$. The goal of the lifelong learner is to find the set of solutions to these subproblems as a set of modules $M = \{m_1, \ldots, m_k\}$, such that learning to solve a new problem reduces to finding how to combine these modules optimally. Each module can be viewed as a processing stage in a hierarchical processing pipeline, or equivalently as functions in a program, and the goal of the agent is to find the correct module to execute at each stage and the instantiation of that module (i.e., its parameters). We formalize the compositional RL problem as a graph $\mathcal{G} = (\mathcal{V}, \mathcal{E})$ (e.g., Figure 1) whose nodes

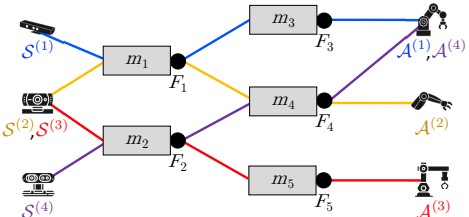

Figure 1: Compositional RL problem graph.

are the subproblem solutions along with the state and action spaces: $\mathcal{V} = \mathcal{F} \bigcup \check{\mathcal{S}} \bigcup \check{\mathcal{A}}$, where $\check{\mathcal{S}} = \text{unique}(\{\mathcal{S}^{(1)}, \ldots, \mathcal{S}^{(T_{\max})}\})$ and $\check{\mathcal{A}} = \text{unique}(\{\mathcal{A}^{(1)}, \ldots, \mathcal{A}^{(T_{\max})}\})$ are the unique state and action spaces. Each subproblem $F_i$ corresponds to a latent representational space $\mathcal{Y}_i$, generated by the corresponding module $m_i : \mathcal{X}_i \mapsto \mathcal{Y}_i$. For example, an object detection module could map an image to a segmentation mask, to be used by subsequent modules to make action decisions. Similarly, the $\mathcal{S}^{(t)}$'s and $\mathcal{A}^{(t)}$'s can serve as representation spaces $(\mathcal{X}_t, \mathcal{Y}_t)$.

Problem $\mathcal{Z}^{(t)}$ is then specified as a pair of state and action nodes $(\mathcal{S}^{(t)}, \mathcal{A}^{(t)})$ in the graph, and the goal is to find a path between those nodes corresponding to a policy $\pi^{(t)*}$ that maximizes $R^{(t)}$. More generally, the graph formalism allows for recurrent computations via walks with cycles, and parallel computations via concurrent multipaths; an extended definition of *multiwalks* trivially captures both settings. In this work, we consider the path formulation, and we restrict the number of edges in the graph by organizing the modules into layers, as explained in Section 5.1. This formalism is related to that of Chang et al. (2019) for the supervised compositional setting, but is adapted to RL. From this definition, we outline two concrete problem settings, based on the amount of information available to the agent. Both settings show how learning could benefit from compositional solutions.

**Zero-shot generalization with full information**     In some scenarios, the agent may have access to a task descriptor that encodes how the current task relates to others in terms of their compositional structures. This descriptor might be sufficient to combine modules into a solution (i.e., zero-shot generalization), provided that the agent has learned to map the descriptors into a solution structure. The descriptor could take different forms, such as a multi-hot encoding of the various components, natural language, or highlighting the target objects in the input image. Our experiments study multi-hot descriptors as a means to provide the compositional structure. Formally, we assume that the descriptor is given as an external input $t \in \mathcal{T}$, and that there exists some function $s : \mathcal{T} \times \mathcal{M} \mapsto \Pi$ that can map this input and an optimal set of modules $M$ into an optimal policy for the current task $s(t, M) = \pi^{(t)*}$. This would enable the agent to achieve *compositional* generalization: the ability to solve a new task entirely by reusing components learned from previous tasks.

**Fast adaptation with restricted information**     In other scenarios, the agent is not given the luxury of information about the compositional structure. This is common in RL, where the only supervisory signal is typically the reward. In this case, the agent would be incapable of zero-shot transfer. Instead, we measure generalization as its ability to *learn from limited experience* a task-specific function $s^{(t)} : \mathcal{M} \mapsto \Pi$ that combines existing modules into the optimal policy for the current task $s^{(t)}(M) = \pi^{(t)*}$. Intuitively, if the agent has accumulated a useful set of modules $M$, then we would expect it to be capable of quickly discovering how to combine and adapt them to solve new tasks.

## 4   EXAMPLE COMPOSITIONAL GENERALIZATION TASKS FOR RL

We now define two domains that intuitively exhibit the properties outlined in Section 3. More details, including the observation and action spaces and the reward, are provided in Appendix B.

**Discrete 2-D world**     The agent's goal is to reach a specific target in an environment populated with static objects that have different effects. The tasks are compositional in three hierarchical levels:
- *Agent dynamics*: We artificially create four different dynamics, each corresponding to a permutation of the actions (e.g., the `turn_left` action moves the agent forward, `turn_right` rotates left). For each task, the effect of the agent's actions is determined by the chosen dynamics.
- *Static objects*: We introduce a chain of static objects in each task, with a single gap cell. `Walls` block the agent's movement; the agent must open a `door` in the gap cell to move to the other side. `Floor` cells have an object indicator, but have no effect on the agent. `Food` gives a small positive reward if picked up. Finally, `lava` gives a small negative reward and terminates the episode.
- *Target objects*: There are four colors of targets; each task involves reaching a specific color.

There are 64 tasks of the form "reach the COLOR target with dynamics N interacting with OBJECT." If the agent has learned to "reach the red target with dynamics 0" and "reach the green target with dynamics 1", then it should be able to "reach the red target with dynamics 1" by recombining its knowledge. Tasks were simulated on `gym-minigrid` (Chevalier-Boisvert et al., 2018).

These 2-D tasks capture the notion of functional composition we study in this work, and prove to be notoriously difficult for existing lifelong RL methods. This demonstrates both the difficulty of the problem of knowledge composition and the plausibility of neural composition as a solution. To

show the real-world applicability of the problem, we propose a second domain of different robot arms performing a variety of manipulation tasks that vary in three hierarchical levels.

**Robot manipulation**    We consider four popular commercial robotic arms with seven degrees of freedom (7-DoF): Rethink Robotics' Sawyer, KUKA's IIWA, Kinova's Gen3, and Franka's Panda.

- *Robot dynamics*: These arms have varying kinematic configurations, joint limits, and torque limits, requiring specialized policies. All dynamics are simulated in `robosuite` (Zhu et al., 2020).
- *Objects*: The robot must grasp and lift a can, a milk carton, a cereal box, or a loaf of bread. The varying sizes and shapes imply that no common strategy can manipulate all these objects.
- *Obstacles*: The workspace the robot must act in may be free (i.e., no obstacle), blocked by a wall the robot needs to circumvent, or limited by a door frame that the robot must traverse.

Each task is one of the 48 combinations of the above elements, just like in the 2-D case. Intuitively, if the agent has learned to manipulate the milk carton with the IIWA arm and the cereal box with the Panda arm, then it could recombine knowledge to manipulate the milk carton with the Panda arm.

## 5    Proposed approach for modular lifelong RL

We now describe our framework for learning modular policies for compositional RL tasks. At a high level, the agent constructs a different neural net policy for every task by selecting from a set of available modules. The modules themselves are used to accelerate the learning of each new task and are then automatically improved by the agent with new knowledge from this latest task.

### 5.1    Neural modular policy architecture

We propose to handle modular RL problems via neural composition. Modular architectures have been used in the supervised setting to handle compositional problems (Andreas et al., 2016; Rosenbaum et al., 2018). In RL, a few recent works have considered similar architectures, but without a substantial effort to study their applicability to truly compositional problems (Goyal et al., 2021; Yang et al., 2020). One notable exception proposed a specialized modular architecture to handle multi-task, multi-robot problems (Devin et al., 2017). We take inspiration from this latter architecture to design our own modular architecture for more general compositional RL.

Following the assumptions of Section 3, each neural module $m_i$ in our architecture is in charge of solving one specific subproblem $F_i$ (e.g., finding an object's grasping point in the robot tasks), such that there is a one-to-one and onto mapping from subproblems to modules. All tasks that require solving $F_i$ will share $m_i$. To construct the network for a task, modules are chained in sequence, thereby replicating the graph structure depicted in Figure 1 with neural modules.

A typical modular architecture considers a pure chaining structure, in which the complete input is passed through a sequence of modules. Each module is required to not only process the information needed to solve its subproblem (e.g., the obstacle in the robot examples), but also to pass through information required by subsequent modules. Additionally, the chaining structure induces brittle dependencies among the modules, such that changes to the first module have cascading effects. While in MTL it is viable to learn such complex modules, in the lifelong setting the modules must generalize to unseen combinations with other modules after training on just a few tasks in sequence. One solution is to let each module $m_i$ only receive information needed to solve its subproblem $F_i$, such that it need only output the solution to $F_i$. Our architecture assumes that the state can factor into module-specific components, such that each subproblem $F_i$ requires only access to a subset of the state components and passes only the relevant subset to each module. For example, in the robotics domain, robot modules only receive as input the state components related to the robot state. Equivalently, each element of the state vector is treated as a variable and only the variables necessary for solving each subproblem $F_i$ are fed into $m_i$. This process requires only high-level information about the semantics of the state representation, similar to the architecture of Devin et al. (2017).

At each depth $d$ in our modular net, the agent has access to $k_d$ modules to choose from. Each module is a small neural network that takes as input the module-specific state component along with the output of the module at the previous depth $d-1$. Note that this differs from gating networks in that the modular structure is fixed for each individual task instead of modulated by the state or input. The number of modular layers $d_{\max}$ is the number of subproblems that must be solved (e.g., $d_{\max} = 3$ for (1) grasping an object and (2) avoiding an obstacle with (3) a robot arm).

## 5.2 SEQUENTIAL MODULE LEARNING

Our method for lifelong learning of modular policies accelerates the learning of new tasks by leveraging existing modules, while simultaneously preventing the forgetting of knowledge stored in those modules. We achieve this by separating the learning into two main stages: an online stage in which the agent discovers which modules to use and explores the environment by modifying a copy of those modules, and an off-line stage in which the original modules are updated with data from the new task. The rationale is that, in the early stages of training, there are two conflicting goals: 1) flexibly adapting the module parameters to the current task to explore how to solve it and 2) keeping the module parameters as stable as possible to retain performance on earlier tasks. Our method (Algorithm 1) consists of the following stages.

**Initialization** Finding a good set of initial modules is a major challenge. In order to achieve lifelong transfer to future tasks, we must start from a sufficiently strong and diverse set of modules. For this, we train each new task on disjoint sets of neural modules until all modules have been initialized. Intuitively, this corresponds to the assumption that the first tasks presented to the agent are made up of disjoint sets of components or subproblems, though we show empirically that this assumption is not necessary in practice.

---

**Algorithm 1** Lifelong Compositional RL

$T \leftarrow 0$
**loop**
  $t \leftarrow \texttt{getTask}(); \quad T \leftarrow T + 1$
  **if** $T \leq k$    // Initialize modules
    $\texttt{steps} \leftarrow 0; \quad s_d \leftarrow T \; \forall d$
  **else** // Find module combination
    $s, \texttt{steps} \leftarrow \texttt{discreteSearch}()$
    $\texttt{bckModules} \leftarrow \texttt{clone}(M)$
  $\pi^{(t)}, Q^{(t)} \leftarrow \texttt{modularNets}(M, s)$
  // Online exploration
  **while** $\texttt{steps} < \texttt{onlineSteps}$
    $s, a, r, s' \leftarrow \texttt{rollouts}(\pi^{(t)}, \texttt{iterSteps})$
    $\texttt{buffer}[t].\texttt{push}(s, a, r, s')$
    $\pi^{(t)}, Q^{(t)} \leftarrow \texttt{RLStep}(s, a, r, s', \pi^{(t)}, Q^{(t)})$
    $\texttt{steps} \leftarrow \texttt{steps} + \texttt{iterSteps}$
  // Off-line comp. improvement
  **if** $T < \texttt{numModules}$
    $M \leftarrow \texttt{bckModules}$
  **for** $i = 1, \ldots, \texttt{offlineSteps}$
    **for** $t = 1, \ldots, \texttt{seenTasks}$
      $s, a, r, s' \leftarrow \texttt{buffer}[t].\texttt{sample}()$
      $\pi^{(t)} \leftarrow \arg\min_{\tilde{\pi}} \texttt{NLL}(\tilde{\pi}(s), a)$
      $a' \leftarrow \arg\max_{\tilde{a}:\pi^{(t)}(s', \tilde{a}) > \tau} Q^{(t)}(s', \tilde{a})$
      $\texttt{targetQ} \leftarrow r + Q^{(t)'}(s', a')$
      $Q^{(t)} \leftarrow \arg\min_{\tilde{Q}}(\tilde{Q}(s, a) - \texttt{targetQ})$

---

presented to the agent are made up of disjoint sets of components or subproblems, though we show empirically that this assumption is not necessary in practice.

**Online training of new tasks** In most RL tasks, during the initial stages of training, the agent struggles to find relevant knowledge about the task. Lifelong RL methods typically ignore this, and incorporate new (likely incorrect) information into the shared knowledge all throughout training. Instead, we keep shared modules fixed during this online training phase, subdivided into two stages:

- *Module selection* We consider two versions of our method. In one case, the choice of modules is given to the agent. In the other case, the agent must discover which modules are relevant for the current task. Since a diverse set of modules has already been initialized, this stage uses exhaustive search of the possible module combinations in terms of the reward they yield when combined.
- *Exploration* Once the set of modules to use for the current task has been selected, the agent might be able to perform reasonably well on the task. However, especially on the earliest tasks, it is unlikely that modules that have never been combined will work together perfectly. Therefore, the agent might still need to explore the current task. In order to avoid catastrophic damage to existing neural components and at the same time enable full flexibility for exploring the current task, we execute this exploration via standard RL training on a *copy* of the shared modules.

The rationale for executing selection and exploration separately is that we want the selected modules to be updated as little as possible in the next and final stage. If we were to instead jointly explore via module adaptation and search over module configurations, we are likely to find a solution that makes drastic modifications to the selected modules. If instead we restrict module selection to the fixed modules, then this stage is more likely to find a solution that requires little module adaptation.

**Off-line incorporation of new knowledge via batch RL** While the exploration stage enables learning about the current task, this is typically insufficient for training a lifelong learner, since 1) we would need to store all copies of the modules in order to perform well on new tasks, and 2) the modules obtained from initialization are often suboptimal, limiting their potential for future transfer. For this reason, once the agent has been given enough experience on the current task, newly discovered knowledge is incorporated into the original shared modules. It is crucial that this accommodation step does not discard knowledge from earlier tasks, which is not only necessary for solving those earlier tasks (thereby avoiding catastrophic forgetting) but possibly also for future

tasks (i.e., forward transfer). Drawing from the lifelong learning literature, we use experience replay over the previous tasks to ensure knowledge retention. However, while experience replay has been tremendously successful in the supervised setting, its success in RL has been limited to very short sequences of tasks (Isele & Cosgun, 2018; Rolnick et al., 2019). In part, this limited success stems from the fact that typical RL methods fail in training from fully off-line data, since the mismatch between the off-line data distribution and the data distribution imposed by the updated policy tends to cause degrading performance. For this reason, we propose a novel method for experience replay based on recent batch RL techniques, designed precisely to avoid this issue. Concretely, at this stage the learner uses batch RL over the replayed data from all previous tasks as well as the current task, keeping the selection over components fixed and modifying only the shared modules. Note that this is a general solution that applies beyond compositional algorithms to any lifelong method that includes a stage of incorporating knowledge into a shared repository after online exploration. Appendix E validates that this drastically improves the performance of one non-compositional method.

We use *proximal policy optimization* (PPO) for online training and *batch-constrained Q learning* (BCQ) for batch RL. BCQ simultaneously trains an actor $\pi$ to mimic the behavior in the data and a critic $Q$ to maximize the values of actions that are likely under the actor $\pi$ (Fujimoto et al., 2019b;a). In the lifelong setting, this latter constraint ensures that the Q-values are not over-estimated for states and actions that are not observed during the online stage. Note that other batch RL methods could be used instead. Additional implementation details are included in Appendix C.

## 6 EXPERIMENTAL EVALUATION

Our first evaluation sought to understand the compositional properties of the tasks from Section 4 and the models learned on those tasks, yielding that learned components can be combined in novel ways to solve unseen tasks. This motivated our second evaluation, which explored the benefits of discovering these components in a lifelong setting, showing that our method accelerates the learning of future tasks and avoids forgetting. Evaluation details and ablative tests are in Appendices D and E, and source code is available at: github.com/Lifelong-ML/Mendez2022ModularLifelongRL.

### 6.1 ZERO-SHOT TRANSFER TO UNSEEN DISCRETE 2-D TASK COMBINATIONS VIA MTL

To assess the compositionality of the tasks we constructed, we provided the agent with a fixed graph structure following the formalism of Section 3. This way, the agent knows *a priori* which modules to use for each task and need only learn the module parameters. The architecture uses four modules of each of three types, one for each task component (static object, target object, and agent dynamics). Each task policy contains one module of each type, chained as static object → target object → agent. Modules are convolutional nets whose inputs are the module-specific states and the outputs of the previous modules. Agent modules output both the action and the Q-values. We trained the agent in a multi-task fashion using PPO, collecting data from all tasks at each training step and computing the average gradient across tasks to update the parameters. Since each task uses a single module of each type, gradient updates only affect the relevant modules to each task.

We trained the agent on various sets of discrete 2-D tasks and compared it against two baselines: training a separate single-task (STL) agent on each task, and training a single monolithic network across all tasks in the same multi-task fashion. To ensure a fair comparison, the monolithic MTL network received as input a multi-hot encoding of the components that constituted each task. Figure 2 (left) shows that our method is substantially faster than the baselines by sharing relevant information across tasks.

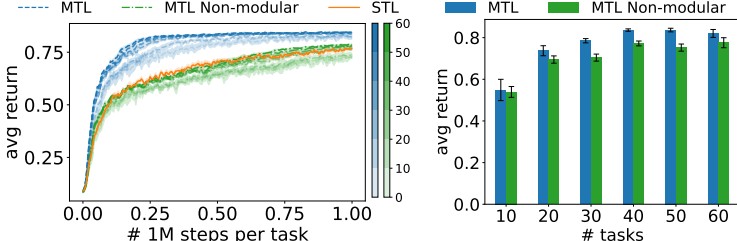

Figure 2: Returns of STL (on 64 tasks) and MTL (on various tasks, per the colorbar) on 2-D tasks. The modular architecture captures the relations across tasks, accelerating learning. Training on more tasks improves results (left). Generalization of pre-trained modules to unseen combinations vs. the number of training tasks (right). Modules can be combined in novel ways to achieve high performance without training. Shaded regions and error bars show std. errors across 6 seeds.

These results suggest that our modular architecture accurately captures the relations across tasks in the form of modules. To verify this, we evaluated the performance of the agent on tasks it had not encountered during training. To construct the network for each task, we again provided the agent with the hard-coded graph structure, but kept the parameters of the modules fixed after MTL training. Figure 2 (right) shows the high zero-shot performance our method achieves, revealing that the modules can be combined in novel ways to solve unseen tasks without any additional training.

## 6.2 Lifelong discovery of modules on discrete 2-D tasks

The results of Figure 2 encourage us to envision a lifelong learning agent improving modules over a sequence of tasks and reusing those modules to learn new tasks much more quickly. In this section, we study the ability of Algorithm 1 to achieve this kind of lifelong composition. We consider two instances of our approach: one in which the agent is given the hard-coded graph structures (Comp.+Struct.) and one in which the agent is given no information about how modules are shared and must therefore discover these relations autonomously via discrete search (Comp.+Search). Once the structure is selected for one particular task, data for that task is collected via PPO training, starting from the parameters of the selected modules. Finally, this collected data is used for incorporating knowledge about the current task into the selected modules via batch RL, using data from the current task and all tasks that reuse any of those modules to avoid forgetting. To match our assumptions, unless otherwise stated the initial tasks presented to the agent contain disjoint sets of components.

We compared against the following baselines: *STL*; *P&C* (Schwarz et al., 2018), a similar method that keeps shared parameters fixed in a first stage, and pushes new knowledge into the shared parameters in a second stage, but uses a monolithic network, making it much harder to find a solution that works for all tasks; *Online EWC* (Schwarz et al., 2018), which continually trains all tasks into a monolithic network with a quadratic penalty for deviating from earlier solutions; and *CLEAR* (Rolnick et al., 2019), which uses replay over previous tasks' trajectories with importance sampling and behavior cloning, but trains a standard monolithic network in a single stage. Lifelong baselines were provided with the ground-truth multi-hot indicator of the task components as part of the input.

Figure 3 (left) shows the average learning curves of the lifelong agents trained on all 64 possible 2-D tasks. Even though tasks are trained sequentially, we show the averaged curves to study the ability to accelerate learning: the fact that the curves for our compositional methods are above STL shows that they achieve forward transfer. Note that this acceleration occurs despite our methods using an order of magnitude fewer trainable parameters than STL ($86, 350$ vs. $1, 080, 320$). Additionally, both our methods improve the modules over time, as demonstrated by the trend of increasing zero-shot performance as more tasks are seen (Figure 3, center). P&C also learns faster than STL, but as we will see, P&C catastrophically forgets how to solve earlier tasks. Other lifelong baselines perform much worse than STL, since they are designed to keep the solutions to later tasks close to those of earlier tasks, which fails in compositional settings where optimal task policies vary drastically.

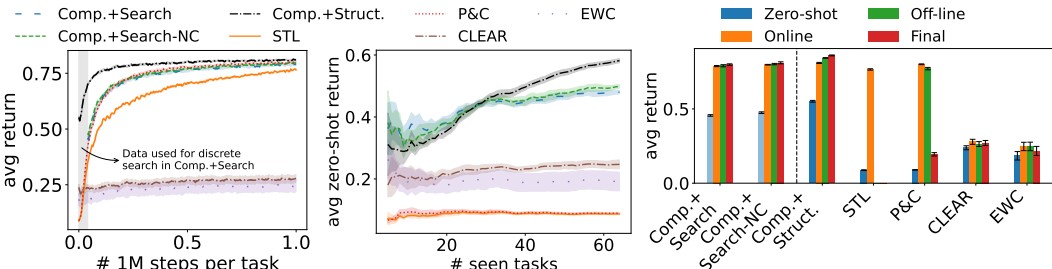

Figure 3: Avg. returns of STL and lifelong agents on 64 compositional 2-D discrete tasks. Our compositional methods accelerate the training w.r.t. STL, demonstrating forward transfer (left). As compositional methods train on more tasks, they improve modules, achieving higher zero-shot performance when combined in novel ways (center). P&C also achieves forward transfer, but it forgets how to solve earlier tasks, while our compositional methods retain performance—Comp.+Struct. even achieves backward transfer (right). Comp.+Search performs better than baselines that receive multi-hot task descriptors. "Zero-shot" for Comp.+Search (shaded) is after discrete search, which does require data. Off-line and Final performances for STL are omitted for clarity, since it is does not perform lifelong training. Shaded regions and error bars represent std. errors across 6 seeds.

In these 2-D tasks, lifelong learners should completely avoid forgetting, since there exist models (compositional and monolithic) that can solve all possible tasks (see Figure 2). Figure 3 (right) shows the average performance as each task progresses through various stages: the beginning (Zero-shot) and end (Online) of online training, the consolidation of knowledge into the shared parameters (for Comp. and P&C only, Off-line), and the evaluation after all tasks had been trained (Final). Our methods are the only that achieve forward transfer without suffering from any forgetting. Moreover, Comp.+Struct. achieves *backward transfer*: improving the earlier tasks' performance after training on future tasks, as indicated by the increase in performance from the Off-line to Final bars.

To study the flexibility of our method, we evaluated it on a random sequence of tasks, without forcing the initial tasks to be composed of distinct components. Figure 3 shows that this lack of curriculum (Comp.+Search-NC) does not hinder the forward or backward transfer of our approach. In addition, Appendix E demonstrates that our method can learn well with different numbers of modules.

### 6.3 LIFELONG DISCOVERY OF MODULES ON REALISTIC ROBOTIC MANIPULATION TASKS

Having validated that our approach achieves forward transfer without forgetting on the 2-D tasks, we carried out an equivalent evaluation on the more complex and realistic robotic manipulation suite. The architecture was similarly constructed by chaining one module for each type of obstacle, object, and robot arm, and each such module was a multi-layer perceptron. We compare against P&C, the best lifelong baseline in the 2-D tasks, and STL. We empirically found that PPO would output overconfident actions in this continuous-action setting (i.e., high-magnitude outputs unaffected by the stochastic action sampling) when initialized from the existing modules directly, which limited our agent's ability to learn proficient policies. Therefore, we applied some modifications to the online PPO training mechanism which permitted successful training of all tasks at the cost of often inhibiting zero-shot transfer (see Appendix D). Figure 4 (left) shows the learning curves of both our compositional methods. All lifelong agents learned noticeably faster than the base STL agent, and compositional methods were fastest, despite using an order of magnitude fewer trainable parameters than STL ($165, 970$ vs. $1, 040, 304$). Figure 4 (right) also shows that the off-line stage led to a decrease in performance. However, like in the 2-D domain, training on subsequent tasks led to backward transfer, partially recovering the performance of the earlier tasks as future tasks were learned. P&C was incapable of retaining knowledge of past tasks, leading to massive catastrophic forgetting.

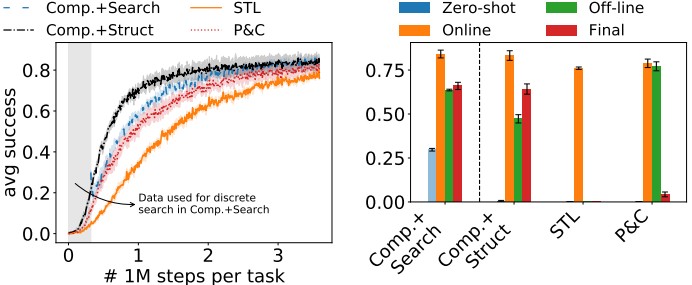

Figure 4: Avg. success of STL and lifelong agents on 48 compositional robot manipulation tasks. Compositional methods again achieve forward transfer (left). The off-line stage causes a drop in performance, but further training the modules on future tasks achieves backward transfer and partially recovers the lost performance (right). Shaded/error bars represent std. errors over 3 seeds.

## 7 CONCLUSIONS AND LIMITATIONS

We formulated the problem of lifelong compositional RL, presenting a graph formalism along with intuitive compositional domains. We developed an algorithm that learns neural compositional models in a lifelong setting, and demonstrated that it is capable of leveraging accumulated components to more quickly learn new tasks without forgetting earlier tasks and while enabling backward transfer. As a core component, we use batch RL as a mechanism to avoid forgetting and show that this is a strong choice for avoiding forgetting more broadly in methods with multi-stage training processes.

One limitation of our work is the scalability with respect to the number of modules, requiring to attempt all possible combinations for the discrete search. While our experiments showed that this is feasible even on relatively long sequences of 64 tasks, specialized heuristics to reduce the search space would be needed if the number of combinations were much larger.

## 8 REPRODUCIBILITY STATEMENT

We have taken a number of steps to guarantee the reproducibility of our results. First, we provide detailed textual descriptions of the compositional environments used for our evaluations in Appendix B and full details of our algorithmic implementation in Appendix C. Next, we describe precisely the experimental setting of our evaluation, including model architectures, baselines, hyper-parameters, and computing resources in Appendix D. In addition to this, we publicly released the source code needed to reproduce our results at: `github.com/Lifelong-ML/Mendez2022ModularLifelongRL`.

### ACKNOWLEDGMENTS

We would like to thank Marcel Hussing for valuable discussions about robotics experiments. We also thank David Kent and the anonymous reviewers for their comprehensive and useful feedback. The research presented in this paper was partially supported by the DARPA Lifelong Learning Machines program under grant FA8750-18-2-0117, the DARPA SAIL-ON program under contract HR001120C0040, the DARPA ShELL program under agreement HR00112190133, and the Army Research Office under MURI grant W911NF20-1-0080.

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

Appendices to
# Modular Lifelong Reinforcement Learning
# via Neural Composition

## A  CONNECTION BETWEEN FUNCTIONALLY COMPOSITIONAL RL AND HIERARCHICAL RL

In Section 2 of the main paper, we briefly discuss the connection between our proposed functionally compositional RL problem and the popular hierarchical RL setting. Given the close connections between the two, in this section we expand upon that discussion.

The primary conceptual difference between hierarchical RL and our proposed functionally compositional RL problem is that hierarchical RL considers composition of sequences of actions *in time*, whereas we consider composition *of functions* that, when combined, form a full policy. In particular, for a given compositional task, all functions that make up its modular policy are utilized at every time step to determine the action to take (given the current state).

Going back to our example of robot programming from Section 1 in the main paper, modules in our compositional RL formulation might correspond to sensor processing drivers, path planners, or robot motor drivers. In programming, at every time step, the sensory input is passed through modules in some pre-programmed sequential order, which finally outputs the motor torques to actuate the robot. Similarly, in compositional RL, the state observation is passed through the different modules, used in combination to execute the agent's actions.

Hierarchical RL takes a complementary approach. Instead, each "module" (e.g., option) is a self-contained policy that receives as input the state observation and outputs an action. Each of these options operates in the environment, for example to reach a particular sub-goal state. Upon termination of an option, the agent selects a different option to execute, starting from the state reached by the previous option. In contrast, our compositional RL framework assumes that a single policy is used to solve a complete task.

An integrated approach is possible that decomposes the problem along both a functional axis and a temporal axis. This would enable selecting a different functionally modular policy at different stages of solving a task, simplifying the amount of information that each module should encode. Consequently, the framework we propose could be used to learn individual options, which would then be composed sequentially. This would enable options to be made up of functionally modular components, simplifying the form of the options themselves and enabling reuse *across* options. Research in this direction could drastically improve the data efficiency of RL approaches.

## B  COMPOSITIONAL ENVIRONMENT DETAILS

In this section, we describe in more detail the environments we propose in Section 4 in the main paper for evaluating functional composition in lifelong RL.

### B.1  DISCRETE 2-D WORLD

Each task in the discrete 2-D world consists of an $8 \times 8$ grid of cells populated with a variety of objects, and is built upon `gym-minigrid`. Below, we describe the core elements of our proposed environment and how they vary according the task components.

**Observation space**  The learner receives a partially observable view of the $7 \times 7$ window in front of it, organized as a $h \times w \times c$ image-like tensor, where $h = w = 7$ are the height and width of the `agent`'s field of view, and $c = 7$ is the number of channels. Each channel corresponds to one of: `wall`, `floor`, `food`, `lava`, `door`, `target`, and `agent`. The first four channels are binary images with ones at locations populated with the relevant objects. The `door` channel contains a zero for any cell without a `door` or with an open `door`, and a one for any cell with a closed `door`. The `target` channel has all-zeros except for locations in which a `target` is located, which are populated with an integer indicator of the color between one and four. Finally, the `agent` channel has all-zeros except for the location of the `agent`, which is populated with an integer indicator of

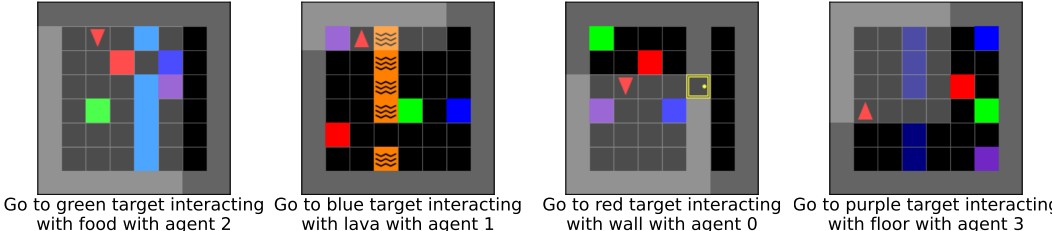

Go to green target interacting with food with agent 2    Go to blue target interacting with lava with agent 1    Go to red target interacting with wall with agent 0    Go to purple target interacting with floor with agent 3

Figure B.1: Visualization of various instantiations of our compositional discrete 2-D tasks. The highlighted area represents the agent's field of view.

the `agent`'s orientation (`right`, `bottom`, `left`, or `up`) between one and four. The `agent` can observe past all objects except `walls` and closed `doors`, which occlude any objects beyond them.

**Action space** Every time step, the agent can execute one of six actions: `turn_left`, `turn_right`, `move_forward`, `pick_object`, `drop_object`, and `open_door`. These discrete actions are deterministic, so that they always have the intended outcome. However, we artificially create distinct dynamics by permuting the ordering of the actions in the following four orders; for readability, we have grayed out actions whose effect stays constant across permutations:

0. `turn_left`, `turn_right`, `move_forward`, `pick_object`, `drop_object`, `open_door`
1. `turn_left`, `turn_right`, `open_door`, `pick_object`, `drop_object`, `move_forward`
2. `turn_left`, `move_forward`, `turn_right`, `pick_object`, `drop_object`, `open_door`
3. `turn_left`, `move_forward`, `open_door`, `pick_object`, `drop_object`, `turn_right`

**Reward function** We primarily rely on the original sparse reward function provided by `gym-minigrid`, which gives a zero at every time step except at the end of a successful episode. The terminal reward value is computed as $1 - 0.9(i/H)$, where $i$ is the time step at which the target is reached and $H$ is the horizon of the environment. In tasks where `food` is present, the agent gets an additional reward of $0.05$ for every piece of food it picks up with the `pick_object` action. In contrast, the agent receives a penalty of $-0.05$ if it steps on a `lava` object.

**Initial conditions** The $8 \times 8$ grid is surrounded by a wall. The initial state of every episode is set by randomly sampling the locations of all objects in the scene. First, a horizontal location $x$ is picked in the range $[2, w - 2]$, where the static object is placed. All cells $i, j$ such that $i = x$ are populated with the task's static object, except for one individual cell at a random vertical location $y : i, j = x, y$. Cell $x, y$ is left empty in all tasks except those whose static object is `wall`, in which case a closed `door` is placed. The agent is placed randomly in some location not occupied by the static object, and facing randomly in any of the four possible directions. Finally, one target object of each of the four possible colors is randomly placed in any remaining free spaces in the environment.

**Episode termination** For all tasks, the episode terminates upon reaching the correct target, or after $H = 64$ time steps, whichever happens first. The episode immediately terminates if the agent steps on a `lava` object.

Figure B.1 shows example tasks created by sampling one component of each type.

### B.2 ROBOTIC MANIPULATION TASKS

Each task in the robotic manipulation domain consists of a single robot arm in a continuous state-action space, with a single object and (optionally) an obstacle. The dynamics of the task are simulated on `robosuite`, and all robots use a general-purpose gripper by Rethink Robotics with two parallel fingers. We now provide details of underlying MDP of tasks within this domain, and how it varies according the task components.

**Observation space** Each time step, the agent is given a rich observation that describes all elements in the task. The robot arm state is described by a 32-dimensional vector, concatenating the sine and cosine of the joint positions, the joint velocities, the end-effector position, the end-effector orientation in unit quaternions, and the gripper fingers' positions and velocities. The target object's state is described by a 14-dimensional vector that concatenates the position and orientation of the object in

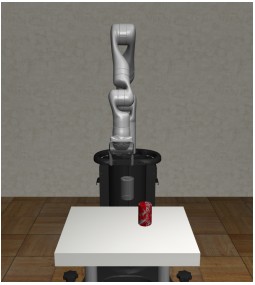 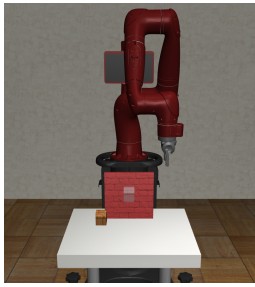 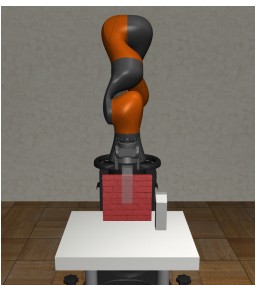 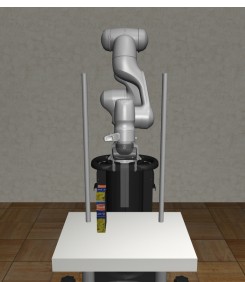

Figure B.2: Visualization of various instantiations of our compositional robotic tasks.

global coordinates, and the position and orientation of the object relative to the end-effector. The obstacle is similarly described with its position and orientation in global and end-effector coordinates. The goal towards which the target is to be lifted is again described with its position and orientation in both coordinate frames, as well as the relative position of the target object with respect to the goal. Note that many of these elements are redundant and in principle unnecessary for solving the task at hand. However, we found that this combination of observations leads to tasks that are much more easily learned by the STL agent.

**Action space**   The agent's actions are continuous-valued, eight-dimensional vectors, indicating the change in each of the seven joint positions and the gripper. For reference, this corresponds to the `JOINT_POSITION` controller in `robosuite`.

**Reward function**   We design dense rewards for our environments similar to those used in `robosuite`. At a high level, the agent receives an increasingly large reward for approaching, grasping, and lifting the object. Additionally, in tasks involving the `wall` obstacle, we found it necessary to provide the agent with additional reward to encourage it to lift the object past the `wall`. Concretely, in tasks with no `wall`, the reward is given by the following piece-wise function:

$$r = \begin{cases} 0.1(1 - \tanh(5d)) & \text{if not grasping} \\ 0.25 + 0.5(1 - \tanh(25h)) & \text{if grasping and not success} \\ 1 & \text{if success ,} \end{cases}$$

and in tasks with `wall`, the reward is given by:

$$r = \begin{cases} 0.05(1 - \tanh(5d_w)) & \text{if not past wall} \\ 0.05 + 0.05(1 - \tanh(5d)) & \text{if past wall and not grasping} \\ 0.25 + 0.5(1 - \tanh(25h)) & \text{if grasping and not success} \\ 1 & \text{if success ,} \end{cases}$$

where $d$ is the distance from the gripper to the center of the object, $h$ is the vertical distance from the object to the target height truncated at 0, and $d_w$ is the $x, z$ distance from the gripper to the wall.

**Initial conditions**   The robot arm is placed at the center of the left edge of a flat table of width $w = 0.39$ and depth $d = 0.49$. The obstacle is placed one quarter of the way from left to right in the table, and at the center. The object location is sampled uniformly at random in the right half of the table, so that the robot must always surpass the obstacle before reaching the object.

**Episode termination**   The episode terminates after reaching the horizon of $H = 500$ time steps.

Figure B.2 shows example tasks created by sampling one component of each type.

## C   ADDITIONAL ALGORITHMIC DETAILS

Our lifelong compositional RL agent faces tasks in sequence, and is always provided with (at least) a task index to indicate the current task being trained. Additionally, our Comp.+Struct. variant is also informed of the graph structure that relates the various tasks.

At a high level, our method uses discrete search during the module selection stage, then trains a copy of the module parameters online via PPO in the exploration stage, and subsequently uses a small

data set of the current and past tasks to train the actual module parameters via BCQ. Implementation details of each of these steps are provided below.

**Online training of new tasks: Module selection**  Upon encountering a new task, the agent selects the best modules to solve the task without any modifications to the module parameters. This is done to ensure that the choice of modules requires minimal modifications to those parameters, as is needed to retain knowledge of the earliest tasks. In our experiments, we perform a discrete search over all possible combinations of parameters, which requires the least amount of additional assumptions. We roll out each of the resulting policies for ten episodes and pick the module combination that yields the highest average return across the ten episodes.

**Online training of new tasks: Exploration**  It is unlikely that the chosen modules will immediately solve the task without modifications, especially for tasks encountered early in the agent's lifetime, when it has not yet learned fully general modules. In the supervised setting, the agent is given a data set that is representative of the data distribution of the current task, which enables the agent to incorporate knowledge into the modules directly using the provided data after module selection (Mendez & Eaton, 2021). However, in RL there is no given data set, and the agent must instead explore the environment to collect data that represents near-optimal behavior. To do this, we execute PPO training, leveraging the selected modules to initialize the policy, but without modifying the actual shared module parameters. BCQ in the next step requires training an action value function $Q$, so for compatibility we modify PPO to compute the state value function ($V$) from the $Q$ function that we actually train. In the discrete setting, we assume a deterministic actor and compute the maximum value from the computed $Q$ values: $V = \max_a Q(s, a)$. In the continuous setting, we instead sample $n = 10$ actions and compute the average $Q$ value across all of them to obtain an approximation of $V$.

**Off-line incorporation of new knowledge via batch RL**  One popular strategy in the supervised setting to incorporate new knowledge into shared parameters is to train parameters with a mix of new and replayed data. In RL, this type of off-line training via standard techniques often leads to degrading performance, due to the mismatch between the original data distribution and the data distribution imposed by the updated policy. Therefore, we use BCQ as a training mechanism that constrains changes to this distribution.

- *Discrete*: In the discrete action setting, the algorithm trains an actor and a critic. The actor is trained to imitate the behavior distribution in the data, so we use the same $\pi$ network as used during PPO training. The critic is trained to compute the $Q$ values, constrained to regions of the data distribution with high probability mass. Concretely, instead of computing $Q(s, a) = r + \gamma \max_{a'} Q(s', a')$ as dictated by the standard Bellman equation, BCQ computes $Q = r + \gamma \max_{a':\pi(s,a')>\tau} Q(s', a')$, where $\tau$ is a threshold below which actions are considered too unlikely under the data distribution. In our experiments, after this stage we evaluate the policy by rolling out actions via Boltzmann sampling of $Q$.

- *Continuous*: The original continuous-action BCQ is substantially more complex, since the actor should mimic a general distribution over the continuous-valued actions, for which BCQ uses a variational autoencoder to represent the arbitrary distribution. However, in our case, we assume that a Gaussian policy can represent the data distribution, since the data was itself generated by a Gaussian actor. Therefore, like in the discrete case, we train the PPO actor $\pi$ to imitate the data distribution. This choice also enables us to drop the perturbation model trained in BCQ to train a separate actor. BCQ trains two separate $Q$ functions in a manner similar to clipped double Q-learning. The target value is computed by

$$r + \gamma \max_{a' \sim \pi(s)} \lambda \min_{j=\{1,2\}} Q_j(s', a') + (1 - \lambda) \max_{j=\{1,2\}} Q_j(s', a') \ ,$$

which we again estimate by sampling $n = 10$ actions from the actor $\pi$. Subsequent evaluations in our experiments are executed by sampling actions from the actor $\pi$.

# D  EXPERIMENTAL SETTING

In this section, we provide additional details about our experimental setting, including a description of our baselines and the selected hyper-parameters. Note that we use our own implementations for all baselines, since there is no open-sourced code available for them. Source code to reproduce our results can be found at: `github.com/Lifelong-ML/Mendez2022ModularLifelongRL`.

### D.1 EVALUATION ON 2-D TASKS

**Model architecture** To construct our architecture for the discrete 2-D tasks, we assume that the underlying graph structure first processes the static object, then the target object, and finally the agent dynamics. Intuitively, the agent's actions require all information about target and static objects, and the agent module should be closest to the output since it directly actuates the agent. The plan for reaching a target object requires

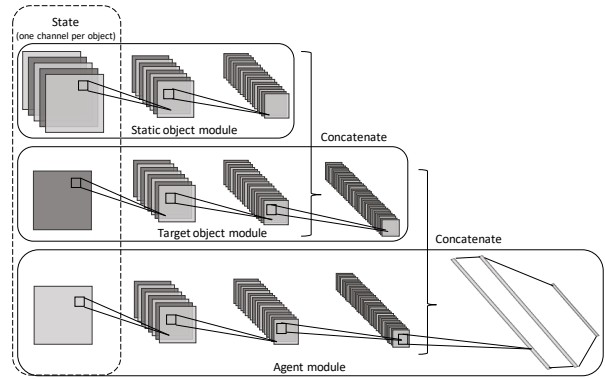

Figure D.3: Modular architecture for discrete 2-D tasks.

information about how to interact with the static object. Interacting with the static object instead could be done without information about the target object.

Static object modules consume as input the five channels corresponding to static objects, and pass those through one convolutional block of $c = 8$ channels, kernel size $k = 2$, ReLU activation, and max pooling of kernel size $k = 2$, and another block of $c = 16$ channels, kernel size $k = 2$, and ReLU activation. Subsequent modules, which require incorporating the outputs of the previous modules, include a pre-processing network that transforms the module-specific state into a representation compatible with the previous module's output, and then concatenate the two representations. The concatenated representation is then passed through a post-processing network to compute a representation of the overall state up to that point. Target object modules pre-process the target object channel with a block with the same architecture as static object modules, concatenate the pre-processed state with the static object module's output, and pass this through a single convolutional block of $c = 32$ channels, kernel size $k = 2$, and ReLU activation. Similarly, agent modules pass the agent channel through a pre-processing net with the same architecture of the target object module (minus the concatenation with the static object module output), concatenate the pre-processed state with the output of the target object module, and pass this through separate multi-layer perceptrons for the actor and the critic with a single hidden layer of $n = 64$ hidden units and $\tanh$ activation. The learner then constructs a separate architecture for each task by combining one module of each type. The architecture is shown in Figure D.3.

**Baselines** We used the following baselines to validate performance.

- *STL* trains a separate model for each task via PPO. Each such model follows the architecture described above, but uses a single module of each type. We chose to use this architecture instead of a standard architecture with the input passed entirely into the first module to ensure a fair comparison, as it yielded substantially better results in our initial evaluations.
- *P&C* uses a similar architecture to that of STL for its knowledge base (KB) and active columns, with additional lateral connections from the KB to the active column. To enable P&C to distinguish among tasks, we add a multi-hot indicator of the task components as inputs to the first fully-connected layer. In the original P&C algorithm, the *progress* phase trains the active column online and the *compress* phase trains the KB column online as well, but imposing an EWC penalty on the shared parameters. We slightly modify this to make P&C closer to our algorithm: the compress phase instead uses replayed data from the current task to distill knowledge from the active column into the KB. By doing this only over the latest portion of the data, this closely matches the P&C formulation that generates data online with the distribution imposed by the (fixed) active policy. This also enables P&C to leverage the entirety of the online interactions for the progress phase, instead of trading off which portion to use for progressing and which portion to use for compressing. We use PPO to train the active column parameters during the progress phase.
- *EWC* uses the same architecture as STL, but again with a multi-hot indicator to discern the different tasks. Throughout its training, EWC modifies all parameters via PPO with an additional quadratic penalty that encourages parameters to stay close to the solutions of the earlier tasks.
- *CLEAR* trains the same architecture as EWC, and also modifies all parameters throughout the training process. In order to prevent forgetting, for every sample collected online and used to compute the PPO loss, $\eta$ samples are replayed from previous tasks to compute the custom CLEAR loss, which balances an importance sampling policy gradient objective (V-trace) and an imitation objective. We ensure replayed samples are evenly split across all previously seen tasks.

Table D.1: Summary of optimized hyper-parameters used by our method and the baselines.

| Algorithm | Hyper-parameter | Discrete 2-D world | Robotic manipulation |
|---|---|---|---|
| PPO | # env. steps | $1M$ | $3.6M$ |
| | # env. steps / update | 4,096 | 8,000 |
| | learning rate | $1e{-}3$ | $1e{-}3$ |
| | minibatch size | 256 | 8,000 |
| | epochs per update | 4 | 80 |
| | $\lambda$ (GAE) | 0.95 | 0.97 |
| | $\gamma$ (MDP) | 0.99 | 0.995 |
| | entropy coefficient | 0.5 | — |
| | Gaussian policy variance | — | fixed |
| | # parameters | $1,080,320$ | $1,040,304$ |
| Compositional | # modules per depth | 4 | 4 |
| | # rollouts / module comb. | 10 | 10 |
| | # replay samples / task | $100,000$ | $100,000$ |
| | # BCQ epochs | 10 | 100 |
| | # parameters | $86,350$ | $165,970$ |
| P&C | $\lambda$ | 10 | 10 |
| | $\gamma$ (EWC) | 1 | 1 |
| | # replay samples / task | $100,000$ | $100,000$ |
| | # distillation epochs | 10 | 100 |
| | # parameters | $70,282$ | $104,384$ |
| EWC | $\lambda$ | $10,000$ | — |
| | $\gamma$ (EWC) | 1 | — |
| | # parameters | $18,928$ | — |
| CLEAR | $\eta$ | 1.5 | — |
| | # parameters | $18,928$ | — |

**Hyper-parameters** We tuned the STL hyper-parameters via grid-search over the learning rate (from $\{1e{-}6, 3e{-}6, 1e{-}5, 3e{-}5, 1e{-}4, 3e{-}4, 1e{-}3, 3e{-}3, 1e{-}2, 3e{-}2\}$) and the number of environment interactions per training step (from $\{256, 512, 1024, 2048, 4096, 8192\}$). Since the static object affects the difficulty of the task, we sampled one task with each static object (`wall`, `floor`, `food`, and `lava`) for each hyper-parameter combination, and computed the average performance as the value to optimize. We reused the obtained PPO hyper-parameters for all lifelong agents. For lifelong agents, we tuned their main hyper-parameter by training on five tasks over five random seeds. For P&C and EWC, we searched for the regularization $\lambda$ in $\{1e{-}3, 1e{-}2, 1e{-}1, 1e0, 1e1, 1e2, 1e3, 1e4\}$. For CLEAR, we searched for the replay ratio $\eta$ in $\{0.25, 0.5, 0.75, 1, 1.25, 1.5, 1.75, 2\}$. Table D.1 summarizes the obtained hyper-parameters.

**Computing resources** Our discrete 2-D experiments were carried out on two small development machines, each with two GeForce® GTX 1080 Ti GPUs, eight-core Intel® Core™ i7-7700K CPUs, and 64GB of RAM. When running two parallel experiments on each machine, our compositional method took an average of approximately one day of wall-clock time to train on all 64 tasks, including a full evaluation of all previously seen tasks after training on each task.

## D.2 EVALUATION ON ROBOT MANIPULATION

**Model architecture** Following the successful results on the discrete 2-D domain, which demonstrated that our chosen architecture correctly captured the underlying structure of the tasks, we manually designed a graph structure for the robotics domain that closely matches

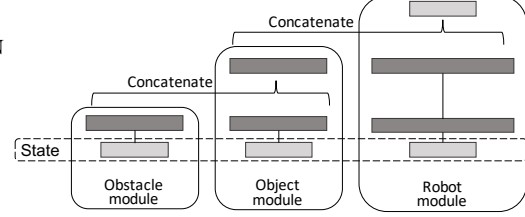

Figure D.4: Modular architecture for robot tasks.

that of the discrete 2-D setting. In particular, we assume that the obstacle is processed first, the object next, and the robot last. The obstacle module passes the obstacle state through a single hidden layer of $n = 32$ hidden units with $\tanh$ activation. The object module pre-processes the object state with another $\tanh$ layer of $n = 32$ nodes, concatenates the pre-processed state with the output of the obstacle module, and passes this through another $\tanh$ layer with $n = 32$ units. Finally, the

robot module takes the robot and goal states as input, processes those with two hidden $\mathtt{tanh}$ layers of $n = 64$ hidden units each, concatenates this with the output of the object module, and passes the output through a linear output layer. Following standard practice for continuous control with PPO, we use completely separate networks for the actor and the critic (instead of sharing the early layers). However, we do enforce that the graph structure is the same for both networks. Moreover, for the critic, the action is fed as an input to the robot module along with the robot and goal states. The architecture is shown in Figure D.4.

**Hyper-parameters**   We did no formal hyper-parameter tuning for this setting. Instead, we created the tasks ensuring that STL performed well. Once the tasks were fixed, we perturbed key PPO hyper-parameters (e.g., the learning rate) and verified that the changes led to decreased performance. We maintained the PPO hyper-parameters fixed for lifelong training. We did tune the regularization coefficient $\lambda$ for P&C by training on five tasks over three random seeds, varying $\lambda$ in $\{1e{-}3, 1e{-}2, 1e{-}1, 1e0, 1e1, 1e2, 1e3, 1e4\}$. Table D.1 summarizes the hyper-parameters used for our robotic manipulation experiments.

**Modifications to the exploration stage**   We found empirically that the lifelong compositional agent would often suffer from premature convergence in robotics experiments, ceasing to explore the space early on in the training. Consequently, we incorporated three modifications to the base PPO algorithm. The first change was to downscale the output layers of the policy and critic networks by a factor of $0.01$ whenever the initial policy achieved no success; this ensured that, if the policy was not close to solving the task, the agent would not be following a highly (and incorrectly) specialized policy, while still leveraging the compositional representations at lower layers. The second modification was to apply a $\mathtt{tanh}$ activation to the last layer of the policy network to limit the magnitude of the outputs, which ensured that stochastic sampling was effective—otherwise, actions would be clipped by the robot simulator and the agent would act deterministically regardless of the variance of the policy. The final adaptation was to fix the variance of the policy to a constant value of $\sigma^2 = 1$ $(\log(\sigma) = 0)$ for the seven joint actions and $\sigma^2 = 1/e$ $(\log(\sigma) = -0.5)$ for the gripper action, since we found that entropy regularization was leading the agent to maximize the variance of irrelevant joint actions while making relevant joints nearly deterministic.

**Computing resources**   Experiments on robotic manipulation tasks were more computationally intensive, due to the cost of physics simulation. We ran these evaluations on an internal cluster, requiring 40 cores and 80GB of RAM per experiment (without GPUs). Our compositional method required approximately two days of wall-clock time to train on all 48 tasks, again including a full evaluation of all past tasks after completing the training of each new task.

# E   ADDITIONAL EXPERIMENTAL RESULTS

One natural question that arises when evaluating methods with various algorithmic and architectural building blocks is, which of these constituent parts are crucial for the obtained performance? In this section, we empirically validate our design choices.

We first verify that the modules required to solve the tasks we design are diverse. Results from Section 6.1 in the main paper show that the discrete 2-D tasks are truly compositional: if the modules are learned well, they can be recombined and reused to solve unseen tasks. However, it is possible that some of these components are essentially the same. For example, perhaps the module for learning to reach the green target could be replaced with the module to reach the red target. This would severely limit the usefulness of our evaluations. As a sanity check, we evaluate the effect of using the incorrect module for evaluation on a task, separated by type of module. Figure E.5-a reveals that using the incorrect static object modules leads to a small (but noticeable) drop in performance, while using the incorrect target object or agent module leads to a drastic drop to nearly random performance. This validates that the modules differ substantially.

We continue by analyzing our architecture choice. A more natural choice of architecture, which we considered early in our development cycle, is a simple module chaining, where the input is passed entirely through a first module, whose output is passed to the next module, and so on. This is in contrast to our architecture, where the input is decomposed into task components and passed separately to distinct modules. We repeated the experiment of Section 6.1 in the main paper with a purely chained architecture and show the results in Figure E.5-b. We found that this chained architecture

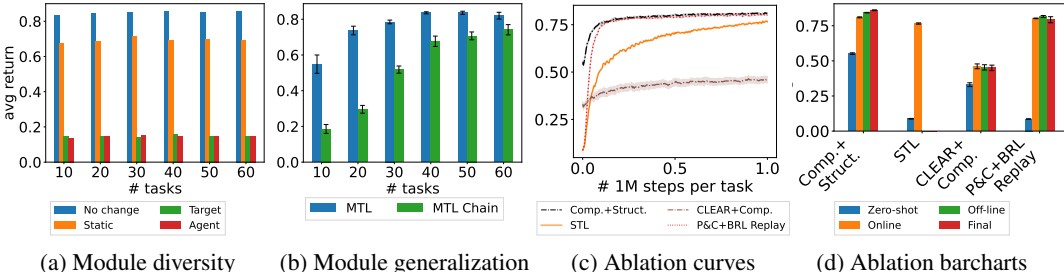

| (a) Module diversity | (b) Module generalization | (c) Ablation curves | (d) Ablation barcharts |

Figure E.5: Ablative analyses on discrete 2-D tasks. (a) Performance of modular MTL agent when constructing the policy with incorrect modules, demonstrating that modules are specialized to solve their assigned subproblem. (b) Generalization to unseen combinations with our proposed modular architecture (MTL) and with a standard chained modular architecture (MTL Chain), showing that modules require far less data to generalize if they are trained on decomposed state representations. (c-d) Performance of our modular architecture trained via CLEAR is much lower than with our method, and our proposed batch RL mechanism to avoid forgetting drastically improves the performance of P&C. Shaded regions and error bars denote standard errors over six random seeds.

cannot generalize nearly as quickly as our modified architecture. This is intuitively reasonable. Consider for example the first module, which is in charge of static object detection. In the chained architecture, this module is further in charge of passing information to subsequent modules about all remaining task components, whereas our architecture need only focus each module on the relevant component, without distractor features from other task components. One alternative view of the same problem is that, to achieve zero-shot generalization, the output of all modules at one depth needs to be compatible with all modules at the next depth. This requires that the output spaces of all modules be compatible. One way to encourage this compatibility is to restrict the inputs to the modules to only the relevant task information, as we do.

To test how our method performs if the architecture uses more or fewer modules of each type than task components, we repeated the experiments on the 2-D domain with varying numbers of neural components, using completely random task sequences (i.e., no curriculum). Figure E.6 shows that our method is remarkably insensitive to the number of modules, performing well with a range of choices.

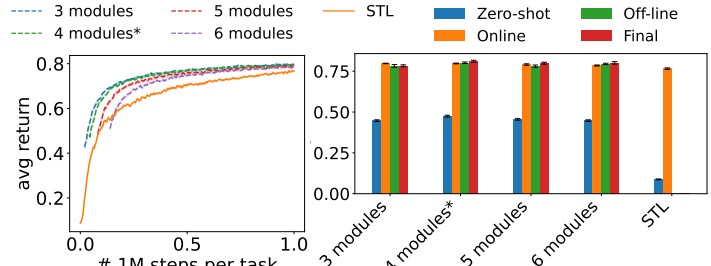

Figure E.6: Avg. returns of Comp.+Search-NC with varying number of modules on 2-D discrete tasks. The number of modules has only minor effects on the overall performance of our approach. Shaded regions and error bars represent std. errors across 6 seeds. *4 modules is the original (correct) value from Section 6.2.

So far, we have shown that modular architectures enable improved performance in the compositional tasks we consider. Since all the lifelong baselines we consider use monolithic architectures, one could think that the improved performance of our method might come solely from the use of a better architecture. To verify that this is not the case, we trained our modular architecture with CLEAR. As shown in Figure E.5-c, while the performance of CLEAR indeed improves, it falls substantially short of matching the performance of our approach. Since CLEAR uses an objective that closely mimics batch RL, but does so online during training of new tasks, this highlights that the advantage of our method comes primarily from the separation of the learning process into multiple stages.

Another major contribution of our work is the use of batch RL as a means for avoiding catastrophic forgetting. To analyze the effect of this choice, we train a version of P&C that replaces EWC in its compress stage with batch RL. Notably, as shown in Figure E.5-d, we found that this almost entirely suppressed the effect of forgetting. Moreover, this led to even-better forward transfer of P&C. This result stresses the fact that avoiding forgetting is not only required for retaining performance of earlier tasks, but also for accumulating knowledge that better transfers to future tasks.

We repeated the batch RL ablative test on our robotic manipulation tasks, yielding that the training of P&C is accelerataed beyond that of STL and batch RL avoids forgetting. However, in this harder setting, the monolithic structure is insufficient to express policies for all tasks and therefore Off-line and Final performance are substantially degraded (see Figure E.7).

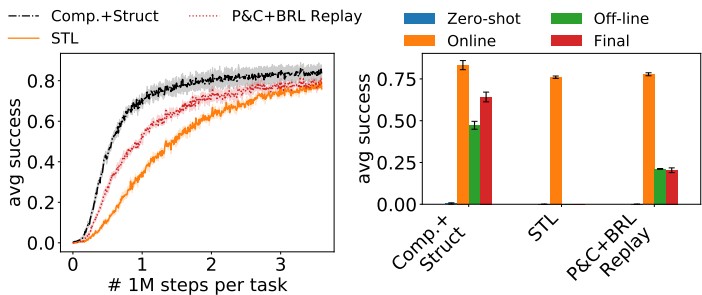

Figure E.7: P&C + batch RL avoids forgetting, but cannot fully incorporate new knowledge due to the monolithic structure.

