# OpenReview forum: "Modular Lifelong Reinforcement Learning via Neural Composition"
_ICLR.cc/2022/Conference — ICLR 2022 Poster_

### Official Review · Reviewer_wUbR · 2021-10-27

**Correctness:** 4
**Technical Novelty And Significance:** 2
**Empirical Novelty And Significance:** 3
**Recommendation:** 8
**Confidence:** 4

**Main Review:**

**Quality, Clarity, Soundness, Correctness**

The paper is well written, the problem is well formulated and introduced, the experiments are clear, and the proposed approach is simple but sound. The benefits of the modular architecture over the non-modular architecture are clear though perhaps less pronounced than expected. Experiments are repeated multiple times to produce results of statistical significance, and the appendix gives all the hyper-parameters and architecture details to reproduce the experiments. I appreciate that the paper clearly points out the extra amounts of data that go into the module-search procedure, which helps with comparison against the other methods. To the best of my knowledge, the claims made in the paper are supported by the empirical results shown.

My main criticism is that the paper stops short of tackling the most important problems in lifelong compositional RL, which are: (i) designing the architecture when the correct number of modules is unknown (and the depth, i.e. the number of layers of modules), (ii) training when the correct sequence of modules is unknown and brute-force search is too costly, (iii) training without disjoint initial training tasks. I think each of these problems is hard, and fully solving only one of them would merit at least another publication. But I am afraid that the current paper implements the “baselines” that one would like to see in any attempt to solve said problems, but makes no attempt to do the latter. At least some solution attempts could be taken e.g. from the supervised literature, e.g. Chang 2019 who train a module-selection policy via RL. Another strong simplifying assumption is that the state can be separated (a priori) into relevant sub-parts for each module such that only relevant information is fed to each module - this is perhaps the fourth (iv) important and difficult problem in training compositional architectures.


**Improvements**

1) The main improvement that I would like to see is an attempt to train the architecture without knowledge of the correct module-sequence and without disjoint initial training tasks (the latter might turn out to be not possible, which would also be an interesting finding). This attempt would probably consist of several sub-experiments including ablations and controls (e.g. training with too many modules, etc.). I understand that this is mostly beyond of what’s possible during the rebuttal phase, yet I think this would be by far the biggest improvement of the current manuscript.

2) While the plots in Fig 2 (left), Fig 3 (left) and Fig 4 (left) nicely visually summarize the performance, they do not provide information about “how long” it takes for the modular MTL to “overtake” STL. Does MTL need to see many tasks where it gets gradually faster and faster to reach high performance, or is there an initial phase of little improvement followed by a “take-off” where all subsequent tasks are learned rapidly (does that occur after each module has been trained once)? Showing the mean only hides all of these dynamics and it would be nice to see this in (a series?) of appropriate plots. It would also help answer questions like: is modular MTL worth the overhead if it’s only 32 or 16 tasks instead of 64, etc.

3) Figure 2 (right) - the performance gains of modular over monolithic MTL are smaller than intuitively expected. One reason could be that the monolithic MTL successfully makes use of the correct module-sequence which is fed to the monolithic architecture as well. To test for this it would be nice to see performance of both modular and monolithic MTL when feeding in random information for the module-sequence to use. Prediction: modular MTL without search should catastrophically deteriorate monolithic MTL should deteriorate if it uses this information but should remain the same if the information is not used.

**Comments**

4) Another control experiment: “knocking out” individual modules (e.g. reinitializing one module randomly) for the modular MTL with search should lead to deterioration of the corresponding tasks and only these tasks. I have no strong reasons to believe otherwise, yet it would be an interesting control experiment to see whether the modules have really fully specialized, or whether they become responsible for multiple task variations when training with search.


**Summary Of The Paper:**

**Update after authors' response:** After reading the other reviews, the updated manuscript and the authors response I think the authors have made several small improvements based on the reviewers' suggestions. Typically I would now be leaning towards a score of 7. However, this year's conference only allows for either a 6 or an 8. To me personally the paper is still slightly closer to 6 than 8, but to make a clear statement in favor of acceptance and facilitate the reviewers' discussion I have raised my score to an 8. I will clearly state my opinion (and hesitation to strongly endorse acceptance of the paper) during the reviewers' discussion.

**Summary**: The paper proposes a novel method for lifelong compositional RL. The latter is formally introduced in the paper as lifelong RL (learning to solve a continuous stream of tasks without re-visiting previously seen tasks) with families of tasks that have a known compositional structure. The paper proposes to use a modular neural architecture (consisting of layers of modules) to capture the compositional nature of the tasks to learn. Through an appropriate module-selection process the (topologically disjoint) modules become functionally disjoint. This is shown to facilitate forward-transfer (faster learning of novel task variations that follow the compositional structure). The paper investigates two module-selection mechanisms: a fixed mechanism where the correct sequence of modules for each task is known, and a brute-force combinatorial search over all module combination to find the highest scoring combination. Once modules are selected, the selected module-parameters are trained to increase current task-performance. This is then consolidated with previous task experience (in an off-line RL fashion) via batch-RL to avoid catastrophic forgetting. Experiments are shown on a set of 2D grid-world tasks, and a number of simulated robot arms with different sensory setups, and the method performs well against a number of baseline methods.

**Main contributions**
1) Definition of compositional families of RL tasks. As the paper correctly points out, a number of previous work on compositional architectures has used variations of off-the-shelf (lifelong) learning tasks, which are not explicitly compositional and a beneficial composition of modules is often not known intuitively (often not even the number of modules is intuitively clear). The two task-families introduced overcome these issues and are a nice contribution towards more meaningful experimentation. The gridworld task is conceptually easy and does not have very complicated perceptual problem or high-dimensional continuous action-spaces; the robotic-tasks on the other hand is closer to real-world applications (and the corresponding complexities). Significance: I think the tasks are well chosen and might become standard-tasks for lifelong compositional RL.

2) Proposal of a compositional architecture and training procedure. Given that the compositional structure of the tasks is precisely known, it is relatively straightforward to propose an exactly matching architecture. The main innovation is a training procedure to train such an architecture. Unfortunately the paper proposes two fairly straightforward solutions that need strong simplifying assumptions: one, the correct sequence of modules is known and applied accordingly, or two, brute-force search over all module combinations (which becomes exponentially costly with increasing numbers of modules, and requires training-data, and relies on disjoint sets of tasks initially until all modules have been trained at least on one task). Significance: the two module-selection mechanisms serve as important controls and lead to promising results. But they are also fairly straightforward conceptually, and rely on often unrealistically strong assumptions. What I would have liked to see for more impactful results is an attempt to train the architecture from scratch without brute-force search and disjoint tasks initially - one possible attempt would be to use RL to train a module selection policy, similarly to Chang 2019, which is used in the paper as a main inspiration for defining compositional RL tasks.

3) Definition of lifelong compositional RL and using techniques from offline RL to avoid catastrophic forgetting. Significance: these are sensible choices, and thus valid contributions, but both have been conceptually proposed before in slightly different settings (Chang 2019 defines compositional supervised problems and using offline RL to avoid catastrophic forgetting and even have backwards-transfer is also not very far-fetched).


**Summary Of The Review:**

The paper aims at addressing a very timely and important, and notoriously difficult problem: lifelong RL with task-families that have compositional structure. In principle, learning about this compositional structure should greatly facilitate learning-speed for new task-variations (forward-transfer). There are (at least) four main problems to solve: (i) how many modules are needed and what is the required computational depth (i.e. maximum number of sequential module calls), (ii) how are modules selected for a particular training instance (when this is not clear a-priori and brute-force search across module-combinations is too costly), (iii) how are modules trained initially after random initialization (how do they acquire specialization), (iv) how can it be ensured that modules only process relevant information (and thus become invariant to irrelevant variation). The fifth problem, avoiding catastrophic forgetting, is actually solved in the paper by borrowing a technique from batch RL. All of these problems are hard and active areas of research. Admittedly the answer to (iii) might be a curriculum, similar to what's proposed in the paper.

The current paper makes strong simplifying assumptions to reduce the severity of each problem. While this is a sensible starting point to establish baseline results, what I’m missing from the current manuscript is an attempt to tackle (some of) these problems. I am not saying this is trivial, and I would consider it a significant addition to the paper. Without such an addition the paper shows promising and interesting results for situations where these problems do not appear (which is often unrealistic) or for situations where these problems could be solved through some means in the future. The paper is well written, and clear, results are good, and I have no complaints regarding the correctness of results and claims. However, given the strong simplifying assumptions the results are not very surprising, which ultimately limits the impact and significance of the paper. I currently see no reason to reject the paper, but also have a hard time assigning a high score. I think with a bit of extra work (admittedly beyond what’s possible in the rebuttal) the paper could become a landmark piece of work in lifelong compositional RL.

---

> ### Author Response · Authors · 2021-11-15
> **Author response (1/2)**
>
> 1. Suggestions about the approach and experiments
>     1. **Module selection policy**. The reviewer suggests that a better mechanism might follow Chang et al. (2019) and use a module-selection policy. This is a good point to raise. We did consider similar ideas, but note that the problem that we are proposing is slightly different than that of Chang et al. In particular, in Chang et al. there is an implicit assumption that the input contains cues of what the structure of the solution should be. For example, this is evidenced in the symbolic processing tasks: the symbols in the arithmetic expressions can be mapped to corresponding programs/module-sequences. In contrast, we do not assume that there are cues in the input about the compositional structure of the tasks: the agent receives positions of objects (without an indication of which object is present), obstacles (without an indication of which obstacle), and joints (without indication of which robot arm). This means that, for every new task, the agent must explore combinations of modules to find which performs best. This distinction is nicely discussed in Alet et al. (2018) in the supervised setting, who resort to a similar solution to perform discrete search. The particular discrete search strategy employed by Alet et al. is simulated annealing, which could be directly applied to replace our exhaustive search in case the number of modules grows large. Another possibility considered in prior work is to let the selection over modules be soft and train this selection via gradient-based learning. While this could also be used, it is expected that it would lead to less functionally differentiated modules. However, note that the growth is _polynomial_ in the number of modules, and not exponential as the reviewer suggests. Moreover, note that if the number of possible module combinations were much larger, that would implicitly imply that the number of _tasks_ the agent is required to learn is equivalently larger. While this is an exciting direction for future work, the current state of lifelong RL suggests that it is not yet equipped to consider a sequence of tasks that long that our method couldn't handle it. In particular, we are not aware of any works that consider longer task sequences than we do in this work.
>     2. **Train without known module combination and disjoint training tasks**. Note that we _did_ perform such an experiment: Comp+Search-NC in Fig. 3 is exactly this variant. The agent trains disjoint modules on the first few tasks until all modules are initialized, but **the initial tasks are not guaranteed to be disjoint**. However, we do agree that it might be useful to study settings where our architecture doesn't exactly match the compositional structure of the tasks. Note that, while our _architecture_ was designed specifically for these problems, the _algorithm_ itself doesn't assume that the structure is a perfect fit, and in particular the off-line accommodation stage permits adjusting the solution even when this is not the case. To test this, we have conducted additional experiments in our 2D domain where the agent used the _incorrect_ number of modules. Of course, this prohibits the possibility of initialization with disjoint task components / subproblems, so these experiments mimic the Comp.+Search-NC variant of Figure 3, where the task sequence is completely random. The table below shows the results with various numbers of modules (including the original results with 4 modules). **The results show that our algorithm works well even when using a different number of modules, with substantial jumpstart performance (Zero-shot) and better final convergence (Online) than STL, as well as little to no forgetting (Final). This makes it possible to apply our proposed method to domains where there is no high-level domain knowledge to choose the number of modules appropriately.** We have included this test in Appendix E in our revised version. Note that so far we have only been able to complete 2 random seeds (which already show clear trends and little variance), and we are in the process of completing an additional 4 seeds to match the setup in the remainder of the paper.
>
> | Algorithm   | Zero-shot       | Online          | Off-line        | Final           |
> |:------------|:----------------|:----------------|:----------------|:----------------|
> | 3 modules   | 0.4388 ± 0.0036 | 0.7930 ± 0.0002 | 0.7650 ± 0.0103 | 0.7676 ± 0.0039 |
> | 4 modules   | 0.4834 ± 0.0071 | 0.7939 ± 0.0038 | 0.7955 ± 0.0039 | 0.8193 ± 0.0015 |
> | 5 modules   | 0.4433 ± 0.0034 | 0.7939 ± 0.0088 | 0.7900 ± 0.0043 | 0.7876 ± 0.0113 |
> | 6 modules   | 0.4342 ± 0.0011 | 0.7859 ± 0.0045 | 0.7834 ± 0.0041 | 0.7866 ± 0.0015 |
> | STL         | 0.0864 ± 0.0017 | 0.7512 ± 0.0045 |      -----      |      -----      |

---

> > ### Author Response · Authors · 2021-11-15
> > **Author response (2/2)**
> >
> > 2. **Aggregated performance in Fig. 2,3,4**. Note that the experimental setting in Fig. 2 is different than in Fig. 3 and 4. In Fig. 2, the agent trains on all tasks at the same time, so there is no such notion of getting gradually faster over a sequence of tasks. However, to study precisely the question brought up in the review, Fig. 2 does evaluate MTL on varying numbers of tasks, from 10 to 60 (as indicated by the colorbar). The result shows that even if trained on only 10 tasks the agent already is noticeably faster than the STL agent, and that this improvement increases substantially as the agent trains on more tasks. For lifelong methods, indeed Fig. 3 (left) and 4 (left) show an average over tasks that _are not_ learned in parallel. Fig. 3 (center) studies the question of how initial (zero-shot) performance varies as more tasks are learned. Note that the first point, at the left end of the curve, corresponds precisely to the task immediately after all modules have been initialized. This shows that at this point there is already a noticeable improvement in initial performance by combining existing modules. This improvement gets better and better over time as the agent trains on more tasks.
> >
> > 3. **Non-modular MTL results**. Indeed, the results of Figure 2 (right) suggest that the non-modular or monolithic architecture does captures the relations across tasks given the multi-hot component indicator. We are in the process of running the suggested ablation to verify this, but we are not certain if we will have complete results by the end of the discussion period, given computational bottlenecks. However, one notable point is that extracting this kind of knowledge from the multi-hot indicator is possible only in the MTL setting (as studied in Fig. 2), where the agent sees all this information about many tasks in parallel. When moving to the lifelong setting, none of the monolithic lifelong baselines was capable of finding this structure. Part of the challenge is that when training on the first task, the multi-hot indicator seems superfluous (i.e., it is unnecessary to solve the single task) and therefore the network learns to ignore it. Then, these same parameters must be reused by all subsequent tasks, and overcoming this choice to discard the multi-hot indicator is challenging. Instead, the modularity of our architecture makes it possible for our method to easily find the structure relating the multiple tasks. Of all the non-modular methods, the most similar to our approach considered in our evaluation is our own version of P&C with batch RL replay (Fig. E.5 and E.6 in the Appendix). The separation of the learning process into online and off-line stages, along with the ability to train off-line over replayed data with batch RL techniques, enable this variant to leverage the multi-hot indicator much better than any existing non-modular approach, achieving the best performance among such baselines.
> >
> > 4. **Knocking out individual modules**. This is an interesting ablative test, and we will run it if our compute permits during the discussion period. However, note that if one module is randomly initialized _during training_, future tasks will almost certainly not select the now-random module. This will cause other modules to carry the load of more tasks, which will likely cause a (slight) decrease in performance across all tasks. If instead the reviewer is suggesting to randomly initialize the modules _after training_ only for evaluation, then absolutely tasks that don't use the selected module will not be affected. This is by design and there is no need for such a test: tasks that don't use a module have absolutely no connection to the parameters of that module. There is no possibility that the performance of these tasks will deteriorate.

---

> > > ### Author Response · Authors · 2021-11-16
> > > **Potential re-interpretation of reviewer's improvement suggestion #3**
> > >
> > > After carefully considering the reviewer's request for a study with random module sequences, we realized that there were two potential interpretations:
> > > 1. The agent is given random module sequences / task components _during training_ and subsequently evaluated with other random sequences. This is the interpretation that we had originally made and how we had begun the additional experiments (which would have taken too long to finish, as we pointed out in our initial response).
> > > 2. The agent is given random module sequences / task components _only during evaluation_ after having been trained with correct module sequences. After some thought, we are guessing that this interpretation may more likely be what the reviewer meant, but we would welcome the reviewer to clarify in the discussion. If this is the case, we already have this evaluation in Fig. E.5 (a) in the Appendix for the modular architecture, which demonstrates that indeed the modular architecture is using the information about the module sequences. We had also executed this exact evaluation for the non-modular architecture (as the reviewer suggested), but had chosen to omit it because we did not think that it added significant value (given the analysis in our initial response regarding the difference between leveraging this information in the MTL vs. the lifelong setting). The results of this evaluation are included for completeness in the tables below, where each column represents which of the modules was incorrectly given to the agent. The conclusion is that the non-modular architecture _does_ leverage the information about the task components (in particular about the target and agent components). Interestingly, the modular variant is affected by this perturbation similarly for all numbers of tasks, whereas the non-modular variant becomes more and more sensitive to it as more tasks are seen. This shows that the non-modular agent requires seeing more task combinations to discover the underlying modular structure, which validates our claim that this is infeasible in a lifelong setting.
> > >
> > > **MTL Modular**
> > >
> > > |   # tasks |   No change |   Static |   Target |    Agent |
> > > |----------:|------------:|---------:|---------:|---------:|
> > > |        10 |    0.832253 | 0.673494 | 0.148023 | 0.135269 |
> > > |        20 |    0.842742 | 0.687123 | 0.146999 | 0.14502  |
> > > |        30 |    0.852233 | 0.71091  | 0.141277 | 0.151316 |
> > > |        40 |    0.854253 | 0.689583 | 0.158332 | 0.147055 |
> > > |        50 |    0.851379 | 0.695975 | 0.143982 | 0.147132 |
> > > |        60 |    0.856992 | 0.69243  | 0.144867 | 0.145045 |
> > >
> > > **MTL Non-modular**
> > >
> > > |   # tasks |   No change |   Static |   Target |    Agent |
> > > |----------:|------------:|---------:|---------:|---------:|
> > > |        10 |    0.75688  | 0.702873 | 0.456642 | 0.251978 |
> > > |        20 |    0.7474   | 0.73929  | 0.385544 | 0.159904 |
> > > |        30 |    0.78341  | 0.779827 | 0.347426 | 0.147698 |
> > > |        40 |    0.794829 | 0.78785  | 0.320911 | 0.148477 |
> > > |        50 |    0.792532 | 0.786559 | 0.303798 | 0.138166 |
> > > |        60 |    0.801013 | 0.796394 | 0.282201 | 0.129343 |

---

> > > > ### Comment · Reviewer_wUbR · 2021-11-20
> > > > **Sorry for not being clear, and slow to respond.**
> > > >
> > > > What I had in mind was indeed evaluation-only (i.e. option 2 above). More particularly, I was not very concerned that the modular architecture would not use the module information (though it is always good to perform a sanity check, which you have done in the appendix). My main question was: does the monolithic architecture make use and benefit from the module information? So the only experiment to run was, at evaluation-time, provide random module-information to the monolithic architecture and measure whether and how much the performance degrades.
> > > >
> > > > This information helps with answering the following: sometimes the performance gains from the modular architecture compared to the monolithic architecture are smaller than intuitively expected. How much of the observed performance improvements compared to the monolithic architecture are due to having the module information (which is important and non-trivial extra information) and how much performance improvements can be attributed to the modularity of the architecture? By showing modular performance without the "privileged" module info, we can tease apart these two factors.
> > > >
> > > > Again this is not the most important result - but an interesting ablation that might help practitioners decide whether "it's worth it to set up the modular architecture, or simply augment a monolithic architecture with the module (side-) information".

---

> > > > > ### Author Response · Authors · 2021-11-22
> > > > > **Thank you for clarifying**
> > > > >
> > > > > We are glad that our second interpretation was correct. The results provided in our previous response precisely tackle this question: the answer is that the monolithic architecture _does_ leverage the module information (as confirmed by the fact that using the incorrect module information leads to decreased performance). However, incorporating this information into the monolithic architecture takes far more tasks (in the _batch MTL setting_ where we execute this evaluation), which serves as evidence that in the lifelong setting (where the agent sees only _one_ task at a time, as in our later evaluations) it is far more difficult to discover modular information via monolithic architectures.

---

> > > > > > ### Comment · Reviewer_wUbR · 2021-11-27
> > > > > > **Thanks**
> > > > > >
> > > > > > I think this is a valuable control experiment that adds to the number of interesting findings in the paper.

---

> > > ### Comment · Reviewer_wUbR · 2021-11-20
> > > **Response cont'd**
> > >
> > > 2. Aggregated performance: Thanks for clarifying! I missed that the difference between training in parallel on all tasks vs. training sequentially (because of the lifelong RL setting I had assumed tasks were always presented sequentially).
> > >
> > > 3. I will respond on your updated understanding below.
> > >
> > > 4. Knocking out modules. While results are somewhat expected, there is one possibility left to check (as far as I can tell). During Evaluation, when using Search, knock out a single module ("remove it from the modular architecture" such that the search process can never select this module anymore). The search process will still find a best-performing sequence of modules - the expectation is that this leads to significant deterioration of performance for any task. If the contrary were to happen, then some modules would capture multiple functionalities. This might be a more relevant test (during evaluation) when training with too many modules. Having said all of this, I don't think it's an important experiment just a very minor suggestion - hence I listed it as a "comment" not a "major improvement".

---

> > > > ### Author Response · Authors · 2021-11-22
> > > > **Continued discussion**
> > > >
> > > > We thank the reviewer for continuing to engage in discussion.
> > > >
> > > > 2. **Module selection policy**.
> > > >     - We agree with the reviewer about the importance of continued work on improving the module selection mechanism. In particular, learning a selection policy _for the case where the state representation implicitly or explicitly encodes the compositional solution_ (unlike the search variant of our approach) is an exciting direction for future work.
> > > >     - **Polynomial vs. exponential growth**. While we agree that the brute-force search is a limitation of our approach (as we have acknowledged in our submission and response to reviewers), the search process requires exploring $\mathrm{numModules}^{\mathrm{depth}}$ possible combinations, which is polynomial (and not exponential) in the number of modules.
> > > > 4. **Knocking out modules**. We thank the reviewer for clarifying their suggested experiment. We would like to clarify that there is no additional search during the evaluation phase. At this stage, the agent has already discovered the correct module sequence for each task and uses it directly. If one module were removed, then all agents that relied on this module would need to re-search at evaluation time (as the reviewer suggests). We ran a simple experiment for this, as follows. We took our pre-trained Comp.+Search agents on the 2-D domain, and for each task evaluated its performance if its best static, target, or agent module were not available. For this, we ran a search over the remaining modules of each type (effectively finding the second best module of each type for each task). The table below summarizes the average performance of this evaluation. As expected, there is a substantial drop in performance, demonstrating that modules are highly specialized.
> > > >
> > > > |             |   Static |   Target |    Agent |
> > > > |:------------|---------:|---------:|---------:|
> > > > | best        | 0.706383 | 0.701964 | 0.707878 |
> > > > | second best | 0.48929  | 0.228149 | 0.22176  |

---

> > > > > ### Comment · Reviewer_wUbR · 2021-11-27
> > > > > **Thank you for the response and additional results**
> > > > >
> > > > > Thanks for clarifying the polynomial growth of the search process.
> > > > >
> > > > > Big thanks for running the Knock-out experiment (and clarifying that during final evaluation there is no search over modules in the original setting). Even though the experiment is not crucial for the main message of the paper I think it is nice to see the intuitions quantitatively confirmed (modules show high functional specialization).
> > > > >
> > > > > I have updated my review in the meantime and raised my score to clearly indicate that I am in favor of accepting the paper.

---

> > ### Comment · Reviewer_wUbR · 2021-11-20
> > **Thanks for the detailed response**
> >
> > I want to thank the authors for their detailed response, clarifications, and the additional results. Having read the other reviews and the authors' responses, I remain in favor of accepting the paper and to me the paper is now exactly on the threshold between a 'weak accept' and 'accept' (mainly because of the somewhat limited significance that comes by requiring fairly strong assumptions for the method to work well, and show strong benefits over a monolithic baseline). I will revise my review (and finalize my score) soon, but wanted to answer while the authors still have time to respond to some comments/details (sorry for not answering sooner) - having said that, I don't think much discussion is urgently needed.
> >
> > Re: 1 (Module selection policy). I agree with the authors that Chang et al. (2019) also use privileged information (task info in some experiments) or at least make somewhat strong assumptions about "clues" for the right modular structure from the data (but I think overall the assumptions are weaker than in the current paper). While the current work is certainly a good starting point, I still believe that the most significant improvements could be made by a more sophisticated module selection process (that requires fewer assumptions). The two options considered in the paper are: (i) known, correct module sequence (which is often unrealistic) and (ii) brute-force search (which requires additional data, and suffers from scalability issues in general. I think the next natural degree of sophistication would be (iii) a learned module selection policy. I am not saying that the latter is crucially important for the current paper, but I think it is a very important problem in lifelong modular (reinforcement) learning - making an attempt at (iii) would be very nice (but is easily beyond what's possible in a rebuttal; I'm just pointing this out again to justify why I am hesitant to give a very high score to the paper).
> >
> > I am still a bit confused about the polynomial vs. exponential growth, could the authors please clarify? My current understanding is that the search process goes through all possible module sequences (i.e. all combinations of picking one module for each level), which seems like a scaling-bottleneck when considering increasing numbers of modules.
> >
> > Re: Train without known module combination of disjoint initial tasks. Thanks for pointing out the experiment - minor nit: while the initial tasks are not guaranteed to be disjoint, there is still a fairly high "degree of disjointness". I would still expect the method to break down (module collapse) to some degree when training on initial tasks that are guaranteed to be non-disjoint. However, I think this is somewhat obvious from the method itself, and is simply a current limitation of the approach - no need to run any experiments to verify this; especially since the experiment you already have shows some degree of resilience to violating the "disjoint initial tasks assumption".
> >
> > Thanks for the additional experiments with non-matching numbers of modules - they are a nice addition.

---

### Official Review · Reviewer_HvzU · 2021-10-29

**Correctness:** 3
**Technical Novelty And Significance:** 3
**Empirical Novelty And Significance:** 2
**Recommendation:** 6
**Confidence:** 4

**Main Review:**

I very much agree with the authors' goals: lifelong reinforcement learning is an extremely interesting and underexplored area. I also agree that compositional modularity is an extremely promising direction that merits further study by itself. Unfortunately, the paper suffers from several problems in its current version. However, don not think that these problems are insurmountable: the authors should be able to submit an updated version addressing my main issues in time.

#### Strengths
- The topic and chosen solution are each highly important research directions with enormous room for future research.
- The paper is extremely well written. The language is fluent and the overall structure of the paper is solid.
- The algorithm introduced is a well-engineered solution to the outlined problem

#### Weaknesses
- (This is my main point of concern. If it is addressed to my satisfaction, I will increase my score to weak accept.) While all explanations and descriptions throughout the paper are easily understood, they severely lack in detail that is required to put many statements into context. And while most of this detail can be found in the appendix, I cannot recommend a paper where the appendix is essential to understanding core parts of the main paper for acceptance.
- It is always a delicate balance to jointly introduce a new domain and a new approach, as this can quickly create the impression that a paper co-developed an approach and a domain the approach is bound to excel at, diminishing the appeal of both. Unfortunately, this is the case for the current paper: the particular input permutations (changing the color of the goal) and action-space permutations are an extremely good fit to the algorithm and it's targeted modular architecture. One example is that the number of permutations is aligned with the number of modules.
- Consequently, I find the domain a little too simple. Each tasks will decompose perfectly in representation and action space, even for the robotics domain. This is an unrealistic assumption. An additional experiment on something with a less well-defined structure would have been very interesting. This would have allowed to also investigate other interesting properties, like learning more complex rules that could e.g. involve questions of recursivity or similar.
- Throughout the paper, it is implied that composition is a sequential process. This is a useful simplification as it allows routing-like stacking of neural layers. However, I would strongly argue that it only a simplification: true modular compositionality should allow for arbitrary connections (imagine e.g., two perception skills that are required for logical inference, or recursive rules).

#### Detailed remarks
- p3, l8: I don't understand what "a two-layer abstraction" should mean
- p4, top: Mapping the policy onto a graph of sub-skills seems off, as a graph does not allow for parallelisms that may be required for more complex tasks
- section 5: here is where the bulk of details are missing:
  - section 5.1: "each neural module mi in our architecture is in charge of solving one specific subproblem F" what does that mean? that there are as many modules as subproblems?
  - section 5.1: "A better solution is for each module to receive as input only the information relevant to it, such that its output need not characterize any additional information." I have no idea what this means
  - section 5.1: "our architecture assumes that the state is composed of module-specific components" too vague
  - section 5.1: "At each depth d in our modular net" what is dmax in general? How is dmax determined?
  - section 5.2: "while simultaneously preventing the forgetting of knowledge stored in those modules. We achieve this by separating the learning into multiple stages." way too little detail. I do not understand this either
  - section 5.2: "For this, we train each new task on disjoint sets of neural modules until all modules have been initialized." does this mean that the subtasks are also disjoint?
  - section 5.2: "incorporate new (likely incorrect) information" the information may be superfluous or distracting, but it's definitely not "incorrect"
  - section 5.2: "Since a diverse set of modules has already been initialized, we do this via discrete optimization of the reward with respect to the possible combinations of modules." what does this mean? what is discrete optimization?
  - section 5.2: "However, while experience replay has been tremendously successful in the supervised setting, its success in RL has been limited to very short sequences of tasks." please elaborate
  - section 5.2: A module is copied before it is updated to prevent catastrophic forgetting. I did not find anywhere how the updates are later re-introduced into the architecture.
- section 6: I argue that at least a rough explanation of the architecture is essential when introducing results. I want to know: what are the modules? how many are there? how were they chosen? how many parameters do the different approaches have (the existing pairwise comparisons are insufficient). I especially miss an explanation of how the different modules are partitioned and why they were designed that way?

**Summary Of The Paper:**

The paper introduces an algorithm for lifelong reinforcement learning using functional neural composition. The algorithm first maps a new problem onto a composition of previously acquired modules; then, the agent trains/finetunes the selected module combination on the new task; and finally, the agent incorporates this newly acquired information into the existing modules. This algorithm is then evaluated on two domains, one multi-task lifelong gridworld domain and a multi-task lifelong robotics domain. The algorithm produces superior results to several other lifelong learning approaches.

**Summary Of The Review:**

A well chosen topic, with a highly interesting solution.

Strengths:
- good research direction
- paper is extremely well written
- the core of the paper, the introduced algorithm, is a well-engineered solution to the problem

Weaknesses:
- all details relevant to contextualize both the approach and the results are in the appendix
- the paper jointly introduces a domain and an approach, opening it to the critic that the experiments are hand-crafted towards the approach
- the experiments seem too simple
- I disagree with some assumptions behind compositionality

---

> ### Author Response · Authors · 2021-11-15
> **Author response (1/2)**
>
> Thank you for your feedback. We left a response to all reviewers, and are including here specific responses to your comments. Please let us know if there are additional questions that you would like us to address during the remainder of the discussion period.
>
>
> 1. Questions and suggested revisions. We have answered the detailed questions below, and have addressed them in the revised manuscript. Changes are highlighted in blue for clarity. Note that the page limit has _not_ been increased to address reviewers' concerns, and therefore there is limited flexibility to update the draft. **If there are additional details that require further clarifications, please let us know and we will make every effort to address the suggestions.**
>     1. **Two-layer abstraction**, Section 2. State abstraction methods do a form of functional composition: a policy is trained upon the outputs of a state abstraction module. Therefore, this is a two-layer composition or abstraction.
>     2. **Sequential vs parallel composition**, Section 3. This is an excellent point. While we did focus our exposition on sequential composition of modules in a computation chain, the graph structure itself could also be used for e.g., parallel or recursive computation. This would require a slight modification: the policy for a task would no longer be a path through the graph, but instead it could be viewed as a _multiwalk_, which is a simple extension of a _multipath_ (as in, e.g., multipath routing) which allows for parallel routing to _walks_ which further permit cycles required for recursive computations.
>     3. **Each module solves one subproblem**, Section 5.1. This means that there is a one-to-one and onto mapping from subproblems to modules. Note that the agent is not necessarily informed of this mapping (e.g., Comp.+Search) and may instead be required to find this mapping autonomously.
>     4. **Each module only receives relevant information**, Section 5.1. This means that if module m_i is required to solve problem F_i, then only the information required to solve F_i is passed as input to m_i. This allows m_i to output only the solution to F_i, and not any additional information contained in the input that subsequent modules might require.
>     5. **State decomposes into components**, Section 5.1. This means that the state can be separated into components such that each subproblem F_i is a function only of a subset of the components.
>     6. **How is d_max selected**, Section 5.1. d_max is the number of problems that the agent is required to solve when solving a task. For example, the task "grasp the can with the IIWA arm avoiding the wall obstacle" requires devising a plan to grasp the can (1), a plan to avoid the wall (2), and a mapping from the plans to IIWA motor commands (3).
>     7. **Separate learning into stages**, Section 5.2. While details of the stages were already presented in the named paragraphs just below, we have added an initial description of the two main stages to clarify the intuition presented in the first paragraph of Section 5.2.
>     8. **Disjoint modules = disjoint subtasks?**, Section 5.2. In principle, the assumption behind training disjoint modules is that the subtasks or components are themselves disjoint. This was stated already in the sentence immediately after, but we have added the word "subproblems" explicitly to make it clearer. Note that, in practice, this assumption is not necessary (as shown in Fig. 3 by Comp.+Search-NC).
>     9. **Not incorrect information**, Section 5.2. There are many situations in which the information incorporated by an agent in RL is _incorrect_ and not just superfluous or distracting. For example, when updating a Q-function initially, the target Q-function is unknown (and hence incorrect), and therefore the "label" used to drive the learning of the Q-function is also incorrect. As another example specific to the modular case, if a depth-zero-module has been optimally learned but a depth-one-module is still unrefined, backpropagation might incorrectly penalize the depth-zero-module (even if it is already optimal) for errors of the depth-one-module .
>     10. **What is discrete optimization?**, Section 5.2 We meant to convey that the agent performs exhaustive search over the possible module combinations, selecting the combination that yields the highest performance.
>     11. **Elaborate on limited success of replay in lifelong RL**, Section 5.2. We have added citations to two prominent replay-based RL methods that train on short sequences of tasks, and explained that these methods fail in part due to the mismatch between the data and policy distributions, which we target via batch RL.

---

> > ### Author Response · Authors · 2021-11-15
> > **Author response (2/2)**
> >
> > 1. Continued...
> >     12. **How are updates re-introduced into the architecture?**, Section 5.2. This is done via batch RL. This is stated toward the bottom of page 6: "...once the agent has been given enough experience on the current task, newly discovered knowledge is incorporated into the _original shared_ modules," where we have added the phrase "original shared" to make this clearer.
> >     13. **Architecture explanation**, Section 6. We see the value of adding some details of the architecture choice into the main paper (even though they were already present in the Appendix, as the reviewer pointed out). We have added these details in Sections 6.1 and 6.3. We have additionally added the exact number of parameters of each architecture in the Appendix. Note that while EWC and CLEAR in their original formulations use substantially fewer parameters than our architecture, the ablation of CLEAR with the modular architecture in Figure E.5 in the Appendix demonstrates that the reduced capacity is not the cause for these methods' drastically lower performance.
> >
> > 2. **The problem matches the algorithm**. Indeed, we made the effort to create tasks that evaluate the functional composition we propose. We do this deliberately to demonstrate that the problem itself is indeed realistic and relevant. Especially in the case of robotic experiments, we believe that this shows the promise of our ideas, since the set of tasks, along with their decomposable state and action spaces, are natural in robot learning. While this certainly is not a decomposition that exists for every possible domain, we view the fact that it exists for the highly relevant robotics domain as a strength of our submission. However, we do agree that it might be useful to study settings where our architecture doesn't exactly match the compositional structure of the tasks. Note that, while our _architecture_ was designed specifically for these problems, the _algorithm_ itself doesn't assume that the structure is a perfect fit, and in particular the off-line accommodation stage permits adjusting the solution even when this is not the case. To test this, we have conducted additional experiments in our 2D domain where the agent used the _incorrect_ number of modules. Of course, this prohibits the possibility of initialization with disjoint task components / subproblems, so these experiments mimic the Comp.+Search-NC variant of Figure 3, where the task sequence is completely random. The table below shows the results with various numbers of modules (including the original results with 4 modules). **The results show that our algorithm works well even when using a different number of modules, with substantial jumpstart performance (Zero-shot) and better final convergence (Online) than STL, as well as little to no forgetting (Final). This makes it possible to apply our proposed method to domains where there is no high-level domain knowledge to choose the number of modules appropriately.** We have included this test in Appendix E in our revised version. Note that so far we have only been able to complete 2 random seeds (which already show clear trends and little variance), and we are in the process of completing an additional 4 seeds to match the setup in the remainder of the paper.
> >
> > | Algorithm   | Zero-shot       | Online          | Off-line        | Final           |
> > |:------------|:----------------|:----------------|:----------------|:----------------|
> > | 3 modules   | 0.4388 ± 0.0036 | 0.7930 ± 0.0002 | 0.7650 ± 0.0103 | 0.7676 ± 0.0039 |
> > | 4 modules   | 0.4834 ± 0.0071 | 0.7939 ± 0.0038 | 0.7955 ± 0.0039 | 0.8193 ± 0.0015 |
> > | 5 modules   | 0.4433 ± 0.0034 | 0.7939 ± 0.0088 | 0.7900 ± 0.0043 | 0.7876 ± 0.0113 |
> > | 6 modules   | 0.4342 ± 0.0011 | 0.7859 ± 0.0045 | 0.7834 ± 0.0041 | 0.7866 ± 0.0015 |
> > | STL         | 0.0864 ± 0.0017 | 0.7512 ± 0.0045 |      -----      |      -----      |
> >
> >
> > 3. **Composition as a sequential process**. We agree that other forms of composition should be possible. Our graph formalism supports this more arbitrary form of composition with small modifications (which we have highlighted in the revised draft).

---

> > > ### Comment · Reviewer_HvzU · 2021-11-18
> > > **Good changes to the document; still fall a little short**
> > >
> > > I like the additions to the document. However, I still find that they fall a little short. After re-reading the updated section 5, I can still not say with certainty how the modules relate to the problem exactly. In particular, the added sentence "One solution is to let each module mi only receive the information needed to solve its subproblem Fi , such that it need only output the solution to Fi and not additional information
> > > related to distracting inputs." is not clear to me at all. How are the inputs "separated"?
> > >
> > > I also like that the authors did additional experiments with different number of modules. It would be great if the authors managed to add these very nice results to the paper - I believe the paper would be much stronger with them.
> > >
> > > I have already updated my score, and I will update it further with additional changes.

---

> > > > ### Author Response · Authors · 2021-11-20
> > > > **Continued discussion**
> > > >
> > > > We thank the reviewer for the continued discussion and for increasing their rating of our submission. Below is our answer to the reviewer's questions, which we have also addressed in a new revision to our manuscript.
> > > >
> > > > The intuition for separating inputs is that the input vector can be thought of as a collection of variables: $\mathbf{x}=[x_1, x_2, \ldots, x_m]$. For example, in the robot domain these variables are: $\mathbf{x}=[\mathrm{robotPose}, \mathrm{objectPose}, \mathrm{obstaclePose}]$. Each of the subproblems $F_i$ that make up a task is only a function of a set of variables $\mathbf{x}_i$. For example, the obstacle component of the robot tasks is only a function of the $\mathrm{obstaclePose}$ variables. Therefore, the neural module $m_i$ corresponding to subproblem $F_i$ takes as input only the relevant variables $\mathbf{x}_i$. We directly include this separation in the design of our architecture, which requires only high-level information about the semantic meaning of each element of the state representation (e.g., $x_0 \mapsto$ "robot joint 0 angular position"). However, this separation could also be automatically learned (e.g., via an attention mechanism); we view this as an exciting direction for future work. We have updated our discussion in the penultimate paragraph of Section 5.1 to more clearly reflect this; changes are highlighted in blue.
> > > >
> > > > Note that we have already included the additional experiment's results in Appendix E. We have now further updated our manuscript to refer the reader to these additional results in the last sentence of Section 6.2 (also highlighted in blue).

---

> > > > > ### Comment · Reviewer_HvzU · 2021-11-29
> > > > > **Still not completely convinced**
> > > > >
> > > > > I understand the intuition - in fact, I share them. However, this step - finding a decomposition of a problem - is one of the hardest problems I know. Partially, because ML models to not share our intuitions - so they may indeed learn a decomposition, but one that has no relationship to our human intuitions.
> > > > >
> > > > > Still, I do like that the results are included. And while I would personally argue that those are too important for the appendix, I still find myself convinced to increase my score.

---

### Official Review · Reviewer_77pf · 2021-11-01

**Correctness:** 4
**Technical Novelty And Significance:** 4
**Empirical Novelty And Significance:** 4
**Recommendation:** 6
**Confidence:** 3

**Main Review:**

Overall, I found this to be an interesting paper. The idea of functional compositionality is rather intriguing, and its application to RL and continual learning is original. The authors have motivated why a modular policy should improve continual learning on tasks with compositional structure, and they have validated their approach with a series of compositional tasks, showing that their approach outperforms several alternative approaches to continual learning on a variety of metrics. In particular, they show empirically that their approach demonstrates better forward transfer (with both better zero-shot performance and faster learning on new tasks) and that their approach not only avoids catastrophic forgetting, but that it can actually demonstrate a degree of backward transfer (with training on new tasks improving performance on old tasks).

Methodologically, the primary limitation of their approach is that the computational graph linking modules together needs to be specified in advance. For each graph node, a set of possible modules is defined in advance (although the module parameters still need to be learned), and the agent must determine which of the possibilities to insert at each node. Moreover, as the authors have pointed out, the complexity of finding the right combination of modules can grow combinatorially with the number of possible modules, making it difficult to scale this to a larger number of modules. Another limitation is that the overall framework must be divided into distinct training phases, with an initialisation phase that must give the modules some initial training before they can be deployed in the computational graph. A final limitation is that while the framework shows good performance on compositional gridworld tasks, the empirical performance advantage seem weaker on higher dimensional robotic manipulation tasks. However, in spite of these limitations, the approach is novel, and these are issues that further development of this approach would need to address.

For me, the primary weakness with this paper is that the exposition of functional compositionality in Section 3 could be clearer, and improving this would help the paper reach a wider audience. The framework and how it was applied to RL only became clearer after reading the subsequent sections, particularly Section 5.1 and the algorithmic details in the Supplement. Providing concrete examples may help ground what was otherwise a rather abstract discussion. For instance, it was initially unclear what the module were doing (as I now understand it, they are just stages in a multi-stage hierarchical processing ), and defining them as solutions to abstract subproblems and as functions that map undefined inputs **X** to undefined outputs **Y** did not help. Part of the issue was that I initially understood the modules in set M as being insertable at any node in the computational graph, which they are not. Moreover, it was not clear in this section that the computational graph is pre-defined, as I initially thought the agent could flexibly compose the modules together to form any graph, and part of the problem it needed to solve was to find the correct graph. Relatedly, Algorithm 1 includes many variables and functions whose meaning did not become clear until a second reading of the paper -- for instance, how the discreteSearch() worked, how the modules are composed together, and how the algorithm relates to the graph in Fig 1. Again, I had initially understood the discreteSearch() to be discovering the graph in addition to the modules that should go in the graph node.

There is one part of the algorithm that remains unclear to me. Towards the end of paragraph 1 in Section 6.2, the authors mention batch RL as using "data from the current task and all tasks that reuse those modules to avoid forgetting". By this statement, I read two possibilities: (a) the algorithm considers all tasks that share some (but not all) modules with the current task or (b) the algorithm considers all tasks that use the exact same set of modules as the current task. It'd be helpful if the authors could clarify what is meant here. Additionally, if (a) is meant, then how are the gradients propagated so that catastrophic interference does not occur in the modules that are not shared with the current task. And if (b) is meant, then how are these tasks considered “different” from the current task given that they are using the exact same policy network.

The authors also claim the use of batch RL to overcome catastrophic forgetting as one of their contributions. Yet it is unclear to me how this is novel. Rehearsal methods have previously been used to overcome catastrophic forgetting in lifelong learning, and in the RL context specifically, Rolnick et al (2019) specifically used replay techniques to overcome forgetting.

Finally, it may be worth noting that besides functional compositionality and temporal hRL compositionality, there has also been work, at least in the cognitive sciences, in getting agents to learn compositional internal models of tasks for the purposes of model-based planning (see e.g. Franklin & Frank, 2018 - https://journals.plos.org/ploscompbiol/article?id=10.1371/journal.pcbi.1006116). Indeed, the authors' compositional gridworld task is reminiscent of the compositional task in this paper, which is in turn based off of the compositional "grid-sailing" task in Fermin et al, 2010 (https://www.tandfonline.com/doi/full/10.1080/00222895.2010.526467).

**Summary Of The Paper:**

This paper introduces and explores the use of *functional compositionality* in lifelong reinforcement learning (RL). In contrast to hierarchical reinforcement, which explores temporal compositionality (the chaining of options or subpolicies across time), functional compositionality involves assembling a novel, overall function (in the case of RL, the policy function) by composing together neural modules that perform multi-stage processing of the policy's inputs. Each module can be regarded as a small function that takes some abstract input X and maps it to an abstract output Y -- the initial set of modules will take the state space as the input X, while the final set will take the actions as the output Y. Modules can perform their computations in parallel, with their outputs concatenated together afterwards, or they can be chained together, with the output of one set of modules serving as the input of the next set. In this setup, the continual learning problem consists of two phases: first, the correct set of modules for the task must be selected and composed into the policy; then, the parameters of those neural network modules must be updated from the data generated by the agent. Selecting the incorrect set of modules will lead to poor transfer and catastrophic forgetting (as the parameters of the incorrect neural network modules get overwritten). This modularity can help improve robustness by reducing dependencies between modules -- this is achieved both through parallelism and through the need of downstream modules to generalise to novel combinations of upstream modules. The authors validate their approaches on a set of tasks with compositional structure, which a functionally compositional policy would be able to exploit, as it could substitute out modules according to the components of the current task. They compare their framework against alternative continual learning baselines and show that their framework is able to (i) avoid catastrophic forgetting; (ii) demonstrate better zero-shot transfer to new tasks; and (iii) even exhibit better backward transfer on training tasks.

**Summary Of The Review:**

Overall, I would recommend this paper for acceptance. The application of functional compositionality to RL policies is quite original, and the approach has been validated empirically. But the paper would benefit significantly from greater clarity in its exposition of functional compositionality and in better defining certain variables and functions in Algorithm 1.

---

> ### Author Response · Authors · 2021-11-15
> **Author response (1/2)**
>
> Thank you for your feedback. We left a response to all reviewers, and are including here specific responses to your comments. Please let us know if there are additional questions that you would like us to address during the remainder of the discussion period.
>
> 1. Comments on the methodology of our approach
>     1. **Computational graph pre-specified**. This is a very subtle point. In our formulation, we limit the number of _edges_ in the graph by ordering the modules in layers (i.e., there are k_0 modules to choose from at the 0-th depth, k_1 at the 1-st depth...) This is not full knowledge of the graph structure, but instead a constraint over that graph. Other modular learning works have had similar restrictions (e.g., Fernando et al. 2017, Yang et al. 2020). Note that, methodologically, permitting each module to be selected at each depth is not significantly harder (growing the search from k^d to (dk)^d). The challenge is that the modules would be required to be _even more general_, which is hard to achieve in the lifelong setting. Another (perhaps more minor) challenge is that in order for modules to be available at any depth, it is necessary for their input and output dimensions to match, which is contrary to common architecture design for convolutional nets.
>     2. **Complexity of discrete search**. Indeed, we agree that this is a limitation of our approach. However, note that the growth is _polynomial_ in the number of modules, which is less severe than the term "combinatorial" might suggest. Moreover, note that if the number of possible module combinations were much larger, that would implicitly imply that the number of _tasks_ the agent is required to learn is equivalently larger. While this is an exciting direction for future work, the current state of lifelong RL suggests that it is not yet equipped to consider a sequence of tasks that long that our method couldn't handle it. In particular, we are not aware of any works that consider longer task sequences than we do in this work.
>     3. **Distinct training phase including initialization**. We do not view the distinct training stages as a limitation but rather as a strength of our approach. Splitting the learning process into multiple stages has already been shown to enable superior lifelong performance (Schwarz et al. 2018, Mendez & Eaton 2021). Note that specifically the initialization stage simply replaces the training of the first few tasks, and therefore incurs no additional cost. Moreover, we show in Fig. 3 that this initialization process succeeds even if the tasks presented to the agent are not constructed from disjoint components or subproblems (Comp.+Search-NC). **This means that the agent doesn't receive information about the true structure for any of the tasks, even at initialization**.
>     4. **Performance in robotics domain**. We have included updated results on our robotic manipulation experiments, upon suggestion from Reviewer 52Uy. The updated results are included in the revised version of Fig. 4. The summary of the results is that the Comp.+Search variant of our approach was capable of achieving a significant amount of zero-shot transfer, which leads to substantial acceleration w.r.t. STL. Note that the changes required for these updated results (explained in our response to Reviewer 52Uy) were not conceptual, but simply in minor experimental choices specific to the robotics evaluation.
>
> 2. Exposition in Section 3
>     1. **What is each module is doing**. We appreciate the reviewer's analogy of modules being stages in a multi-stage processing pipeline. We have included this clarification, along with an illustrative example to our revised Section 3 (paragraph 4, changes in blue).
>     2. **Graph is pre-defined**. As explained above in point 1.1, the graph itself is not pre-defined, but instead a constraint is placed on the order of the nodes in the paths through the graph. We have also added this clarification in paragraph 5 of Section 3.
>     3. **Details of Algorithm 1**. The algorithm is written in terms that are somewhat agnostic to the choice of architecture, which is why it doesn't explicitly tie back to the graph in Fig. 1. These connections are described in Sec. 5.1. In particular, the way in which modules are combined is described in the last paragraph of Sec 5.1: "At each depth d in our modular net, the agent has access to k_d modules to choose from. Each module is a small neural network that takes as input the module-specific state component along with the output of the module at the previous depth d-1." Based on the reviewer's suggestions and those of Reviewer HvzU, we have revised these two sections for additional clarity. **If there are additional details that require further clarifications, please let us know and we will make every effort to address the suggestions.**

---

> > ### Author Response · Authors · 2021-11-15
> > **Author response (2/2)**
> >
> > 3. **Which tasks batch-RL is applied to**. It is certainly (a)--tasks that share _some_ modules with the current task-- since, as the reviewer points out, (b)--tasks that share _all_ modules with the current task--would imply that the policies are the same. In order to avoid interference with modules that are _not_ shared with the current task, gradients are only propagated through the modules learned by the current task, which prevents updates to other tasks that are not in consideration during batch RL. The intuition is that the current task can only include information about the modules it uses. Alternatively, one could train all tasks and all modules, but this is more computationally expensive.
> >
> > 4. **Novelty of batch RL for replay**. Indeed replay has been used in the past for lifelong RL. However, no existing method uses batch RL techniques, which are designed specifically for the setting where an agent has no option to retrain on past tasks. In particular, Rolnick et al. (2019) consider the simpler setting in which the agent is allowed to revisit earlier tasks multiple times, which decreases the distributional shift between collected data and updated policy. In the most extreme case in our evaluations, task 0 is updated up to 63 times _without any additional experience_, which is only feasible due to the application of batch RL techniques.
> >
> > 5. Thank you for pointing us to this additional reference. We will add a brief discussion to our updated manuscript soon.

---

> > > ### Author Response · Authors · 2021-11-24
> > > **Has our response addressed your concerns?**
> > >
> > > As we near the end of the discussion period, we would be grateful if the reviewer can confirm whether our response has addressed their concerns, and let us know if any issues remain. To recap our response, we:
> > > - Clarify the subtle distinction between pre-specifying the whole computational graph (as the reviewer suggests) and placing **constraints over the computational graph** (as we do)
> > > - Emphasize our experiments with Comp.+Search-NC, which **do not require disjoint initial training tasks**
> > > - Make slight modifications to our evaluation in our robotics domain which **enable improved zero-shot transfer**
> > > - **Improve our exposition in Section 3**, incorporating a clearer description of the role of each module, including an illustrative example
> > > - Clarify that existing replay-based lifelong RL methods do not use **batch RL** techniques, as we do, which enables training over **long sequences of tasks without revisiting**

---

> > > > ### Comment · Reviewer_77pf · 2021-11-25
> > > > **Thank you for your detailed response**
> > > >
> > > > I'd like to thank the authors for their detailed response, and I apologise for the lateness of my reply. I think the latest changes to the manuscript have addressed most of my concerns. The exposition of functional modularity is clearer now, and I like the alternative interpretation of learning which functions to chain together in a program. The new results showing resilience to the number of modules used is also quite nice.
> > > >
> > > > One issue I still have slight reservations about is the initialisation, which seems necessary for "seeding" the modules so that they can adequately specialise. The authors have shown that the initial tasks do not need to be entirely disjoint, but one still requires careful curation to ensure that the tasks are still *adequately* disjoint. But the tasks you are using are still significantly disjoint, and as wUbR remarked, in the extreme case where all tasks are similar to each other, you do expect the initialisation to fail. My concern is that in more naturalistic settings, where disjointness between tasks may be more ambiguous, the practitioner will still have to carefully handcraft the initialisation tasks to get the algorithm to work well. Ideally, it'd be nice for an algorithm to not depend heavily on handcrafted elements like this.
> > > >
> > > > That being said, I do believe the latest version of the manuscript is an improvement, and I remain in support of its acceptance.

---

### Official Review · Reviewer_52Uy · 2021-11-03

**Correctness:** 4
**Technical Novelty And Significance:** 3
**Empirical Novelty And Significance:** 3
**Recommendation:** 6
**Confidence:** 4

**Main Review:**

This paper describes a method that is sensible and seems to work well.  It is a creative combination of existing ideas.

My most favorite (and, also, in a way, least) feature of this paper is the strategy of instantiating the modular structures for all the tasks (with appropriate sharing and then performing end-to-end experience replay on the whole ensemble simultaneously!  It's a very nice way to get each of the modules to understand its shared "job" in the overall task ensemble.   But it also seems unsatisfying that we have to remember and re-use all the data.

Can you explain a bit more about what the various colors mean in Figure 2 left?  I guess you're making a plot for 3 (or so) different numbers of tasks (e.g., 20, 40, 60)?

Also, the shaded error regions seem surprisingly small.  Is there really so little variance?  What sources of error does a seed control?  The tasks it is trained on?  Initial weights? Is there any randomization in the domain?

Legends for sparse dotted lines are really hard to view.  It would probably be better just to use solid lines of different colors.

The phenomenon of backward transfer is nice, but why is it surprising?  It seems almost inevitable given your training setup.

Minor: "are the only that achieve"

"Therefore, upon training a new task, we down-scaled the actions and Q-values."  This seems like a very blunt instrument for addressing the problem with PPO.  Isn't there something that would let you retain the actual module weight values and therefore some hope of "zero-shot" module recombination?

"...it was still able to learn slightly faster than STL, despite using an order of magnitude fewer trainable parameters."
Having fewer trainable parameters, if they are organized right, *should* make learning go faster!

Again the stderr seems *so* small in figs 3 and 4.  Please be sure to articulate all the possible sources of variance and explain how many times you're varying each one (and be careful about the statistical analysis:  for example, training a network once but testing it 100 times and then reporting a standard error based on N = 100 is somewhere between not very illuminating and actually misleading.  I don't mean to say that's what you're doing---it's just an example of a common error and the kind of thing to watch out for.)

The reward function for the robotics problems is *very* well-shaped---so well, it seems that one could possibly just write a controller that does gradient ascent on R to reach the objective!  I understand that this is the norm for ``RL'' on robotics tasks, but it is very nearly supervised learning.  It could be interesting as a point of comparison to just generate a big dataset of good trajectories for each task using a planner, and do supervised learning (AKA behavioral cloning) just to see how well it works.


**Summary Of The Paper:**

This paper combines insights from modular meta-learning, lifelong learning and RL to arrive at a system that can learn, online, to train modules that can be recombined to solve future tasks without further training. In addition, training that happens offline can retrospectively improve modules that were constructed to solve earlier tasks.   These methods are demonstrated in a pedagogical domain and on some high-dimensional robot-control problems.

**Summary Of The Review:**

This is a nice combination of mostly existing methods with some good insights for how to put them together.   It's not a major contribution, but it is well described and the experiments seem good, and most importantly, I think it is heading in a very good direction.    Lifelong learning and compositional generalization are critical aspects of moving forward to bigger and more interesting problems and so I think it is important to encourage work in this area.

---

> ### Author Response · Authors · 2021-11-15
> **Author response**
>
> Thank you for your feedback. We left a response to all reviewers, and are including here specific responses to your comments. Please let us know if there are additional questions that you would like us to address during the remainder of the discussion period.
>
> 1. **Re-use data**. This kind of experience replay is an ultra-popular strategy for lifelong supervised learning (e.g., Lopez-Paz & Ranzato 2017, Nguyen et al. 2018, Chaudhry et al. 2019). Even though multiple attempts have been made to adapt this to RL, it has enjoyed limited success (Isele & Cosgun 2018, Rolnick et al. 2019). Our use of batch RL as a replay mechanism enables leveraging accumulated data in lifelong RL. Crucially, note that **we explicitly don't store all the data**, but only a limited amount of data for each trained task (about 10% in our experiments, as detailed in Table D.1 in the Appendix). This is similar to what existing supervised methods do.
>
> 2. **Colors in Fig 2 left**. Blue is our modular MTL agent, green is a standard non-modular MTL agent, and orange is a STL agent trained individually on each task. Light colors are fewer tasks and darker colors are more tasks (as the reviewer suggests, and as described in the colorbar and caption): we train the agent on 10, 20, 30, 40, 50, and 60 tasks to show that training on more tasks leads to faster learning. While this is expected, it has been very difficult to achieve with deep RL.
>
> 3. **Random seed and statistical analysis**. We completely agree with the importance of proper statistical analysis, which is why we took great care in our experimental design. The random seed controls the following sources of variance: parameter initialization, task order (and consequently which tasks are seen in the experiments of Fig. 2 which use subsets of the tasks), randomization in the domain (object, agent, and goal placement in 2D domain; object in robotics domain). Each random trial is executed entirely independently from others: once the random seed is set, all randomization in the training and evaluation uses this random seed. We used 6 random seeds for all 2D experiments and 3 for robot experiments. To more clearly convey the details of this process, the pseudocode is:
> ```
> for each random seed:
>     set the seed value
>     randomly initialize network
>     randomly permute the order over tasks
>     for each task in the new order:
>         for each trajectory to collect:
>             randomly place objects in the environment
>             collect trajectory
>             aggregate trajectory data and take gradient step
>         evaluate agent on all seen tasks
> ```
> 4. **Backward transfer**. While it is not surprising that our method achieves backward transfer (because indeed it is set up for it), backward transfer has rarely been shown in lifelong RL. This is in spite of multiple approaches being designed to achieve backward transfer (e.g., Rolnick et al. 2019, Mendez et al. 2020). It is especially interesting that it happens even with the automatically learned structure over the graph, which demonstrates that the automatically discovered structure is appropriate.
>
> 5. **Down-scale actions and Q-values**. We agree with the reviewer that better alternatives might achieve even better performance. With this in mind, we made a minor change to our evaluation that permits this. We first prevented the agent from predicting overconfident actions by applying a $\tanh$ activation to the output of the policy network. This implies that even if the agent predicts the most confident action ($1$ or $-1$), the noise of the policy will permit it to sample lower-valued actions. Then, we added one additional check before choosing to downscale the networks' outputs: if the best module combination achieves some level of success (e.g., 10%) then train from the modules directly; otherwise rescale the outputs as before. The rationale is that without overconfidence, a policy that achieves some amount of success can be used to bootstrap the learning, but a policy with no success might be _worse_ than random (we verified this latter point empirically). **We applied this technique to the search variant of our method, and achieved zero-shot performance in robotics experiments (see revised draft, Fig. 4).** What's interesting is that this modification enables the Comp.+Search variant of our approach to outperform the Comp.+Struct variant for which we constructed a manual graph structure. This demonstrates the power of data-based search for constructing compositional solutions.
>
> 6. **Fewer trainable parameters**. Indeed, properly shared parameters _should_ improve performance, yet this has been remarkably hard to achieve with non-modular architectures in deep RL, especially in the lifelong setting. This is the point we are trying to convey.
>
> 7. **Reward shaping**. It is a great idea to compare against behavior cloning. We considered it, but crafting a planner for each of the 48 tasks turned out to be prohibitively time consuming.

---

> > ### Author Response · Authors · 2021-11-24
> > **Has our response addressed your concerns?**
> >
> > As we near the end of the discussion period, we would be grateful if the reviewer can confirm whether our response has addressed their concerns, and let us know if any issues remain. To recap our response, we:
> > - Clarify that our method **does not reuse all data** from all tasks during training
> > - Describe the statistical analysis used in our experiments using multiple random seeds to control a variety of factors **during training and testing**
> > - Make slight modifications to our evaluation in our robotics domain which **enable improved zero-shot transfer**

---

> > > ### Comment · Reviewer_52Uy · 2021-11-26
> > > **Thanks for your response**
> > >
> > > Yes, thanks for the detailed response.  I am still pretty surprised by the extremely low variance in many of the experimental results, but that doesn't affect my assessment of the paper.

---

### Author Response · Authors · 2021-11-15
**Response to all reviewers**

We would like to thank all four reviewers for their detailed comments. It is incredibly rare these days to receive such comprehensive feedback, and we believe that these comments will improve the quality of our submission. We hope we can engage with all reviewers during the discussion period to reach a consensus about accepting the paper.

We are providing individual responses to each reviewer, and wanted to highlight the main points here for everyone.
- We have revised our paper to address concerns from Reviewers 77pf and HvzU. While details were included from the start, both reviewers found that understanding some aspects of the problem formulation and algorithm required going through the Appendix. We have included revisions to address these concerns throughout Sections 3 and 5, along with other minor revisions throughout the draft. **Changes are highlighted in blue in the revised paper**. Please note that ICLR is not allocating an additional page this year to address reviewer concerns in the final paper, which is why these revisions cannot be more extensive.
- We made one minor change to the evaluation in Fig. 4 as suggested by Reviewer 52Uy, which led to **improved zero-shot performance in the robotics domain**.
- We ran an ablative test to study settings where our architecture doesn't exactly match the compositional structure of the tasks. Note that, while our _architecture_ was designed specifically for these problems, the _algorithm_ itself doesn't assume that the structure is a perfect fit, and in particular the off-line accommodation stage permits adjusting the solution even when this is not the case. To test this, we have conducted additional experiments in our 2D domain where the agent used the _incorrect_ number of modules. Of course, this prohibits the possibility of initialization with disjoint task components / subproblems, so these experiments mimic the Comp.+Search-NC variant of Figure 3, where the task sequence is completely random. The table below shows the results with various numbers of modules (including the original results with 4 modules). **The results show that our algorithm works well even when using a different number of modules, with substantial jumpstart performance (Zero-shot) and better final convergence (Online) than STL, as well as little to no forgetting (Final). This makes it possible to apply our proposed method to domains where there is no high-level domain knowledge to choose the number of modules appropriately.** We have included this test in Appendix E in our revised version. Note that so far we have only been able to complete 2 random seeds (which already show clear trends and little variance), and we are in the process of completing an additional 4 seeds to match the setup in the remainder of the paper.

| Algorithm   | Zero-shot       | Online          | Off-line        | Final           |
|:------------|:----------------|:----------------|:----------------|:----------------|
| 3 modules   | 0.4388 ± 0.0036 | 0.7930 ± 0.0002 | 0.7650 ± 0.0103 | 0.7676 ± 0.0039 |
| 4 modules   | 0.4834 ± 0.0071 | 0.7939 ± 0.0038 | 0.7955 ± 0.0039 | 0.8193 ± 0.0015 |
| 5 modules   | 0.4433 ± 0.0034 | 0.7939 ± 0.0088 | 0.7900 ± 0.0043 | 0.7876 ± 0.0113 |
| 6 modules   | 0.4342 ± 0.0011 | 0.7859 ± 0.0045 | 0.7834 ± 0.0041 | 0.7866 ± 0.0015 |
| STL         | 0.0864 ± 0.0017 | 0.7512 ± 0.0045 |      -----      |      -----      |

---

### Decision · Program_Chairs · 2022-01-20

**Decision:**

Accept (Poster)

**Comment:**

The paper presents a method for compositional task learning in the continual RL setting, by composing and reconfiguring neural modules. The method is evaluated on mini-grids and simulated robot manipulation tasks.

The reviewers agree, and I concur, that the paper proposes an interesting solution to a difficult and important problem. The paper is well presented and would make a good addition to the multi-task continual learning. The reviewers appreciate the authors' responses and the improvement to the manuscript, and in particular the extra experiments with the wrong number of modules.

The final version of the paper should:

- Clarify the explanation of functional modularity
- Move the relevant pieces to the main text.
- See Gur et al., NeurIPS 2021, https://openreview.net/forum?id=CeByDMy0YTL for a definition of learnable compositional tasks via Petri Net formalism.

Reviewers appreciate the extra experiments with the wrong number of modules.